# Identifying robust bias adjustment methods for European extreme precipitation in a multi-model pseudo-reality setting

Torben Schmith[1], Peter Thejll[1], Peter Berg[2], Fredrik Boberg[1], Ole Bøssing Christensen[1], Bo Christiansen[1], Jens Hesselbjerg Christensen[1,3,4], Marianne Sloth Madsen[1], Christian Steger[5]

[1] Danish Meteorological Institute, Copenhagen , Denmark
[2] Swedish Meteorological and Hydrological Institute, Hydrology Research Unit, Norrköping, Sweden
[3] Physics of Ice, Climate and Earth, Niels Bohr Institute, University of Copenhagen, Copenhagen, Denmark
[4] NORCE Norwegian Research Centre, Bjerknes Centre for Climate Research, Bergen, Norway
[5] Deutscher Wetterdienst, Offenbach, Germany

Correspondence: Torben Schmith (ts@dmi.dk)

## Abstract

Severe precipitation events occur rarely and are often localized in space and of short duration; but they are important for societal managing of infrastructure. Therefore, there is a demand for estimating future changes in the statistics of occurrence of these rare events. These are often projected using data from Regional Climate Model (RCM) simulations combined with extreme value analysis to obtain selected return levels of precipitation intensity. However, due to imperfections in the formulation of the physical parameterizations in the RCMs, the simulated present-day climate usually has biases relative to observations; these biases can be in the mean and/or in the higher moments. Therefore, the RCM results are adjusted to account for these deficiencies. However, this does not guarantee that adjusted projected results will match future reality better, since the bias may not be stationary in a changing climate. In the present work we evaluate different adjustment techniques in a changing climate. This is done in an inter-model cross-validation setup, in which each model simulation in turn plays the role of pseudo-observations, against which the remaining model simulations are adjusted and validated. The study uses hourly data from historical and RCP8.5 scenario runs from 19 model simulations from the EURO-CORDEX ensemble at 0.11° resolution. Fields of return levels for selected return periods are calculated  for hourly and daily time scales based on 25 years long time slices representing present-day (1981-2005) and end-21[st]-century (2075-2099). The adjustment techniques applied to the return levels are based on extreme value analysis and include climate factor and quantile mapping approaches. Generally, we find that future return levels can be improved by adjustment, compared to obtaining them from raw scenario model data. The performance of the different methods depends on the time scale considered. On hourly time scale, the climate factor approach performs better than the quantile mapping approaches. On daily time scale, the superior approach is to simply deduce future return levels from pseudo-observations and the second best choice is using the quantile mapping approaches. These results are found in all European sub-regions considered. Applying the inter-model cross-validation against model ensemble medians instead of individual models does not change overall conclusions much.

# 1   Introduction

Severe precipitation events occur typically either as stratiform precipitation of moderate intensity or as intense localized cloudbursts lasting up to a few hours only. Such extreme events may cause flooding with the risk of loss of life and damage to infrastructure. It is expected that future changes in the radiative forcing from greenhouse gases and other forcing agents will influence the large scale atmospheric conditions, such as air mass humidity, vertical stability, the formation of convective systems, and typical low pressure tracks. Therefore also the statistics of the occurrence of severe precipitation events will most likely change.

Global climate models (GCMs) are the main tool for estimating future climate conditions. A GCM is a global representation of the atmosphere, the ocean and the land surface, and the interaction between these components. The GCM is then forced with observed greenhouse gas concentrations, atmospheric compositions, land use, etc.  to represent the past and present climate, and with stipulated scenarios of future concentrations of radiative forcing agents to represent the future climate.

Present state-of-the art GCMs from the Coupled Model Intercomparison Project Phase 5 (CMIP5, Taylor et al., 2012) and the recent Coupled Model Intercomparison Project Phase 6 (CMIP6, Eyring et al., 2016) typically have a grid spacing of around 100 km or even more. This resolution is too coarse to describe the effect of regional and local features, such as mountains, coast lines and lakes and to adequately describe convective precipitation systems (Eggert et al., 2015). To model the processes on smaller spatial scales, dynamical downscaling is applied. Here, the atmospheric and surface fields from a GCM simulation are used as boundary conditions for a regional climate model (RCM) over a smaller region with a much finer grid spacing, at present typically around 10 km or even less.

An alternative to dynamical downscaling is statistical downscaling. Here large-scale circulation patterns (e.g. the North Atlantic Oscillation) are related to small-scale variables, such as precipitation mean at a station.  One assumes that the large-scale circulation pattern is modelled well by the GCM and therefore the approach is called perfect prognosis. Using the relationship with the small-scale variables,  calibrated on observations, one can obtain modelled local-scale variables (present-day and future) from the modelled large-scale patterns. A recent overview of these methods and validation of them can be found in Gutiérrez et al. (2019).

The ability of present-day RCMs to reproduce observed extreme precipitation statistics on daily and sub-daily time scales is essential and has been of concern. Earlier studies analysing this topic have mostly focused on a particular country, probably due to the lack of sub-daily observational data covering larger regions, such as e.g. Europe. Thus, Hanel and Buishand (2010), Kendon et al. (2014), Olsson et al. (2015) and Sunyer et al. (2017) studied daily and hourly extreme precipitation in different European countries and reached similar conclusions: first that the bias of extreme statistics decreases with smaller grid spacing of the model, and second that extreme statistics for 24 h duration are satisfactorily simulated with a grid spacing of 10 km, while 1 h extreme statistics exhibits substantial biases even at this resolution. Recently, Berg et al. (2019) evaluated high resolution RCMs from the EURO-CORDEX ensemble (Jacob et al., 2014) also used here and reached similar conclusions for several countries across Europe: RCMs underestimate hourly extremes and give an erroneous spatial distribution.

Extreme convective precipitation of short duration is thus one of the more challenging phenomena to
represent physically accurate in RCMs. The reason is that convective events take place on a spatial scale
comparable to the RCM grid spacing of presently around 10 km. Therefore, the convective plumes cannot
be directly modelled. Instead, the effects of convection are parametrised, i.e. modelled as processes on
larger spatial scales (Arakawa, 2004). Thus, the inability to reproduce these short duration extremes can be
explained by the imperfect parametrization of sub-grid scale convection (Prein et al., 2015), which generally
leads to too early onset of convective rainfall in the diurnal cycle and subsequent dampening of the build-
up of convective available potential energy  (Trenberth et al., 2003).
Thus, even RCMs with their small grid spacing may exhibit systematic biases for variables related to
convective precipitation. If there is a substantial bias, we should consider *adjusting* for this in a statistical
sense before any further data analysis. Such adjustment techniques are thoroughly discussed, including
requirements and limitations, in Maraun (2016) and Maraun et al. (2017). There are basically two main
adjustment approaches. In the *delta-change* approach, a transformation is established from the present to
the future climate in the model run. This transformation is then applied to the observations to get the
projected future climate. In the *bias correction* approach, a transformation is established from present
model climate data to the observed climate and this transformation is then applied to the future model
climate to obtain the projected future climate.
Both adjustment approaches come in several flavours. In the simplest one, the transformation consists of
an adjustment of the mean, in the case of precipitation by multiplying the mean by a factor. In the more
elaborate flavour, the transformation is defined by quantile mapping, preserving also the higher moments.
Quantile mapping can use either empirical quantiles or analytical distribution functions. The ability of
quantile mapping to reduce bias has been demonstrated for daily precipitation in present-day climate using
observations, which are split into calibration and validation samples (Piani et al., 2010; Themeßl et al.,
111  2011).
Bias adjustment techniques originate in the field of weather and ocean forecast modelling, where they are
known as model output statistics (MOS). Here output from a forecast model is adjusted for model
deficiencies and local features not explicitly resolved by the model. Applying similar adjustment techniques
to climate model simulations, however, has a complication not present in forecast applications: Climate
models are set up and tuned to present-day conditions and verified against observations, but then applied
to future changed conditions without any possibility to directly verify the model's performance under these
conditions. Therefore, showing that bias adjustment works for present-day climate is a necessary but not
sufficient condition for the adjustment to work in the changed climate.
A central concept of adjustment methods is the assumption of *stationarity* of the bias. For bias correction
this means that the transformation from model to observations is unchanged from the present-day climate
to the future climate, while for delta-change the transformation from present-day climate to future climate
is unchanged from model to observations. In the ideal case of stationarity being fulfilled, the adjustment
methods will work perfectly and produce perfect future projections. If stationarity is not fulfilled,
adjustment may improve projections, or in the worst cases they may degrade projections, compared to
using raw model output. We also note that the adjustment methods themselves may influence the climate
change signal of the model, depending on the bias and the method used (Berg et al., 2012; Haerter et al.,
2011; Themeßl et al., 2012).

Stationarity has been debated in recent years in the literature (e.g. Boberg and Christensen, 2012; Buser et
al., 2010). Kerkhoff et al. (2014) review and discuss two hypotheses: 1) constant bias: unchanged between
present-day and future (i.e. stationarity) and 2) constant relation: bias varies linearly with the signal. Van
Schaeybroeck and Vannitsem (2016) used a pseudo-reality setting with a simplified model and found large
changes in the bias between present-day and future for many variables and violation of both constant bias
and constant relation hypothesis. Chen et al. (2015) concluded that precipitation bias is clearly non-
stationary over North America in that variations in bias is comparable to the climate change signal.
Velázquez et al. (2015) used a pseudo-reality setting involving two models and concluded that constancy of
bias was violated for both precipitation and temperature on monthly time scale. Hui et al. (2019) used a
pseudo-reality setting with GCMs and found significant non-stationarity of bias for annual and seasonal
temperatures. Besides, they point to a large effect on non-stationarity from internal variability.

To thoroughly validate adjustment methods, both a calibration dataset and an independent dataset for
validation are needed. There are two different approaches to obtain this. In split-sample testing, the
observations are divided into calibration and validation parts, often in the form of a cross-validation (e.g.
Gudmundsson et al., 2012; Li et al., 2017a, 2017b; Refsgaard et al., 2014; Themeßl et al., 2011). A variant is
differential split-sample testing (Klemeš, 1986), where the split in calibration/and validation parts is based
on climatological factors, such as wet and dry years, encompassing climate changes and variations into the
validation.

An alternative approach, which we use here, is *inter-model cross-validation*, as pursued by Maraun (2012),
Räisänen and Räty (2013) and Räty et al. (2014) and others. The rationale is here that the members in a
multi-model ensemble of simulations represent different descriptions of physics of the climate system, with
each of them being not too far from the real climate system. Thus, one member of the ensemble
alternatively plays the role of *pseudo-observations*, against which the remaining adjusted models are
validated. Thus, the trick is that we know both present and future pseudo-observations.

The advantage of inter-model cross-validation, is that the adjustment methods are calibrated under
present-day conditions and validated under future climatic conditions. Therefore, it embraces modelled
physical changes between present and future climate, as for instance a shift in the ratio between stratiform
and convective precipitation. In this respect it is a more realistic setting than validation based on split-
sample test. Also, model and pseudo-observations have the same spatial scale, thus avoiding comparing
pointwise observations with area-averaged model data, as is done in the split-sample testing. On the other
hand, the method assumes that the modelled present-day is not too different from observations. If this is
violated, the method will give too optimistic error estimates compared to what can be expected in the real
World. Please cf. also further discussion in Section 5.2.

Inter-model cross-validation has been applied on daily precipitation to evaluate different adjustment methods (Räty et al., 2014). Here we apply a similar methodology European-wide to extreme precipitation on hourly and daily time scales. This has been made possible with the advent of the EURO-CORDEX, a large ensemble of high-resolution RCM simulations with precipitation at hourly time-resolution. Being more specific, we apply the standard extreme value analysis to the ensemble of model data for present-day and end-21$^{st}$-century conditions to estimate return levels for daily and hourly duration. Then we will apply inter-model cross validation on these return levels in order to address the following questions:

1. Do adjusted return levels perform better, according to the inter-model cross-validation, than using raw model data from scenario simulations?
2. Is there any difference in performance between different adjustment methods?
3. Are there systematic differences in point 1 and 2, depending on the daily and hourly duration?
4. Are there regional differences across Europe in the performance of the different adjustment methods?

Giving qualified answers to these questions can serve as important guidelines for analysis procedures for obtaining future extreme precipitation characteristics.

The rest of the paper contains a description of the EURO-CORDEX data (Section 2) and a description of methods used (Section 3). Then follow the results (Section 4), a discussion of these (Section 5) and finally conclusions (Section 6).

# 2   The EURO-CORDEX data

The model simulations used here have been performed within the framework of EURO-CORDEX (Jacob et al. (2014) ; http://euro-cordex.net ), which is an international effort aimed at providing RCM climate simulations for a specific European region (see Figure 1) in two standard resolutions with a grid spacing of 0.44° (EUR-44, ~50 km) and 0.11° (EUR-11, ~12.5 km), respectively. All GCM simulations driving the RCMs follow the CMIP5 protocol  (Taylor et al., 2012) and are forced with historical forcing for the years 1850-2005 followed by the RCP8.5 scenario for the years 2006-2100 (until 2099 only for HadGEM-ES).

We analyse precipitation data in hourly time-resolution from 19 different GCM-RCM combinations from the EUR-11 simulations shown in Table 1 and we analyse two 25 year long time slices from each of these simulations: a present-day time slice (years 1981-2005) and an end-21$^{st}$-century time slice (years 2075-2099).

All GCM-RCM combinations we use are represented by one realization only, and therefore the data material used represents 19 different possible realisations of climate model physics, though acknowledging that some GCMs/RCMs might originate from the same or similar model code and therefore may not be fully independent. The EURO-CORDEX ensemble includes a few simulations, which do not use the standard EUR-11 grid. These were not included in the analysis, since they should have been re-gridded to the EUR-11 grid which would dampen extreme events, thus introducing an unnecessary error source.

Table 1. Overview of the 19 EURO-CORDEX GCM-RCM combinations used. The rows show the GCMs while the columns
show the RCMs. The full names of the RCMs are SMHI-RCA4, CLMcom-CCLM4-8-17, KNMI-RACMO22E, DMI-HIRHAM5,
MPI-CSC-REMO2009 and CLMcom-ETH-COSMO-crCLIM-v1-1. Each GCM-RCM combination used is represented by a
number (1, 3 or 12) indicating which realization of the GCM is used for the particular simulation.

| GCM \ RCM | RCA | CCLM | RACMO | HIRHAM | REMO | COSMO |
|---|---|---|---|---|---|---|
| ICHEC-EC-EARTH | r12 | | r1 | r3 | | |
| MOHC-HadGEM2-ES | r1 | | r1 | r1 | | |
| CNRM-CERFACS-CNRM-CM5 | r1 | | | r1 | | |
| MPI-M-MPI-ESM-LR | r1 | r2 | | r1 | r1 | r1 |
| IPSL-IPSL-CM5A-MR | r1 | | | | | |
| NCC-NorESM1-M | r1 | | | r1 | | r1 |
| CCCma-CanESM2 | | r1 | | | | |
| MIROC-MIROC5 | | r1 | | | | |


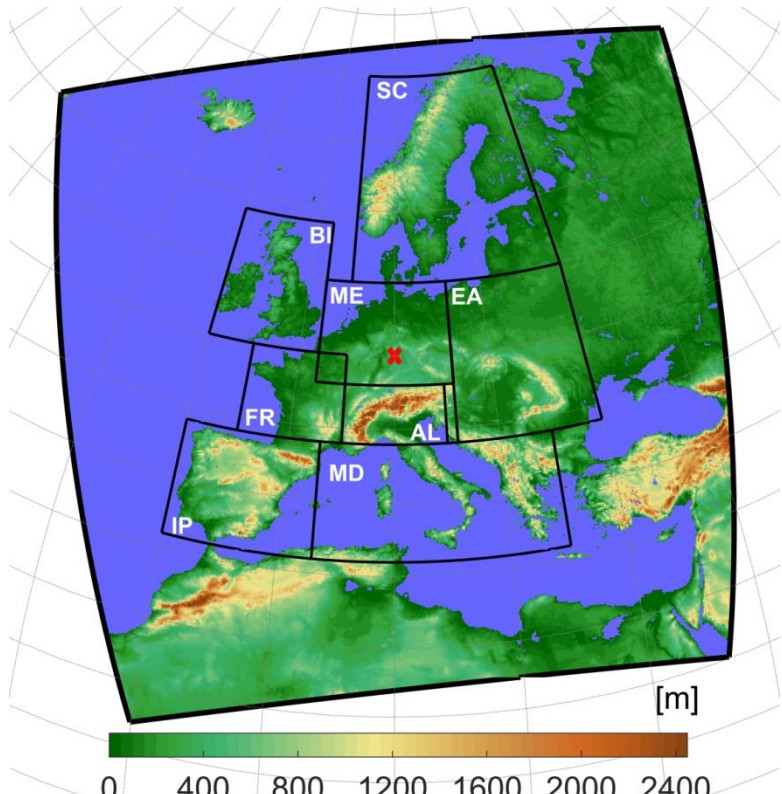

Figure 1. Map showing the EURO-CORDEX region (outer frame) with elevation in colours. PRUDENCE sub-regions (Christensen and
Christensen, 2007) used in the analysis are also shown: BI = British Isles, IP = Iberian Peninsula, FR = France, ME = Mid-Europe, SC =
Scandinavia, AL = Alps, MD = Mediterranean, EA = Eastern Europe. Red cross marks point used in Figure 4.

Generally, GCM results are quite comparable to reality, and many validation studies of GCMs exist, also
with an eye on Europe (e.g. McSweeney et al., 2015). We are aware of the use in some papers of selection
procedures for selecting how to choose sub-sets of available GCMs (e.g. McSweeney et al., 2015; Rowell,
2019). There is, however, no simple quality index that can be generally applied. Any discrimination of GCMs
depends on area, season, and the meteorological field and property being investigated (Gleckler et al.,
2008; e.g. their Fig. 9). Furthermore, these tests and selection procedures are based on subjective
criteria and come with major caveats that impact the uncertainty range largely (Madsen et al., 2017). We
therefore choose, in accordance with most other similar studies, to use an 'ensemble of opportunity' for
the present study.

# 3 Methods

## 3.1 Duration

Extreme precipitation statistics are often described as a function of the time scale involved as intensity-
duration-frequency or depth-duration-frequency curves (e.g. Overeem et al., 2008). We consider two time
scales or *durations*. One is a duration of 1 h, which is simply the time series of hourly precipitation sums
available in each RCM grid point. The other is a duration of 24 h, where a 24 h sum is calculated in a sliding
window with a one hour time step. We will refer to these as hourly and daily duration, respectively. Our
daily duration corresponds to the traditional climatological practice of reporting daily sums but allows
heavy precipitation events to occur over two consecutive days. We also emphasize that the duration, as
defined here, is not the actual length of precipitation events in the model data, but is merely a concept to
define time scales.

## 3.2 Extreme value analysis

Extreme value analysis (EVA) provides methodologies to estimate high quantiles of a statistical distribution
from observations. The theory relies on fundamental convergence properties of time series of extreme
events; for details we refer to Coles (2001).
There are two main methodologies in EVA to obtain estimates of the high percentiles and the
corresponding return levels. In the *classical*, or *block maxima*, method, a generalised extreme value
distribution is fitted to the series of maxima over a time block, usually a year. Alternatively, in the *peak-
over-threshold* (POT) or *partial-duration-series* method, which is used here, all peaks with maximum above
a (high) threshold, $x_0$, are considered. The peaks are assumed to occur independently at an average rate
per year of $\lambda_0$. To ensure independence between peaks, a minimum time separation between peaks is
specified. Theory tells us, that when the threshold goes to infinity, the distribution of the exceedances
above the threshold, $x - x_0$, converges to a generalised Pareto distribution, whose cumulative distribution
function is

$$\mathcal{G}(x - x_0) = 1 - \left(1 + \xi \frac{x - x_0}{\sigma}\right)^{-\frac{1}{\xi}}, x > x_0$$

The parameter $\sigma$ is the scale and is a measure of the width of the distribution. The parameter $\xi$ is the shape
and describes the character of the upper tail of the GPD-distribution; $\xi > 0$ implies a heavy tail which
usually is the case for extreme precipitation events, while $\xi < 0$ implies a thin tail. Note that, quite
confusingly, an alternative sign convention of $\xi$ occurs in the literature (e.g. Hosking and Wallis, 1987).
If we now consider an arbitrary level $x$ with $x > x_0$, the average number of exceedances per year of $x$ will
be

$$\lambda_x = \lambda_0 \left[1 - \mathcal{G}(x - x_0)\right]. \qquad (1)$$


The $T$-year return level, $x_T$, is defined as the precipitation intensity which is exceeded on average once
every $T$ years

$$\lambda_{x_T} T = 1$$

and by combining with (1) we get an expression for the return level $x_T$

$$\lambda_0 [1 - \mathcal{G}(x_T - x_0)]T = 1,$$

from which
$$x_T = \mathcal{G}^{-1}\left(1 - \frac{1}{\lambda_0 T}\right) + x_0. \qquad (2)$$



Data points to be included in the POT analysis can be selected in two different ways. Either the threshold $x_0$
is specified and $\lambda_0$ is then a parameter to be determined or, alternatively, $\lambda_0$ is specified and $x_0$ determined
as a parameter. We choose the latter approach, since it is most convenient when working with data from
many different model simulations.

Choosing $\lambda_0$ is a point to consider: a too high value would include too few data points in the estimation and
a too low value implies the risk that the exceedances $x_T - x_0$ cannot be considered as GPD-distributed. We
choose $\lambda_0 = 3$ in accordance with Berg et al. (2019), which gives 75 data points for estimation for the 25
years long time slices. Hosking and Wallis (1987) investigated the estimation of parameters of the GPD-
distribution and based on this warn against using the often applied maximum likelihood estimation for a
sample size below 500. Instead, they recommend probability-weighted moments and we have followed this
advice here.

We required a minimum of 3 and 24 h separation between peaks for 1 and 24 h duration, respectively. This
is in accordance with Berg et al. (2019) and furthermore, synoptic experience tells us that this will ensure
that neighbouring peaks are from independent weather systems. We found only a weak influence of these
choices on the results of our analysis.

In practical applications of EVA the parameters are estimated with large uncertainties due to limited length
of the time series. The threshold has the smallest relative uncertainty, the scale has a larger relative
uncertainty, and the shape has the largest relative uncertainty. Therefore, also the relative uncertainty of
the return levels increase with increasing $T$, as can be seen from Eq. 2.

## 3.3 Bias adjustments and extreme value analysis
The delta-change and bias correction approaches were introduced in general terms in Section 1. Now we
will formulate EVA-based analytical quantile mapping based versions of the two approaches. In what
follows $O_T$ is the $T$-year return levels estimated from present-day pseudo-observations, while $C_T$ (control)
and $S_T$ (scenario) denote the corresponding return levels, estimated from present-day and end-21[st]-century
model data, respectively. Finally, $P_T$ (projection) denotes the end-21[st]-century return level after bias-
adjustment has been applied.

### 3.3.1 Climate factor on the return levels (FAC)

The simplest adjustment approach is to assume a climate factor on the return level (FAC)

$$P_T = \underbrace{S_T/C_T}_{\substack{Delta-change \\ climate\ factor}} \cdot O_T = \underbrace{O_T/C_T}_{\substack{Bias\ correction \\ climate\ factor}} \cdot S_T$$


We note that the delta-change and bias correction approach are identical for the FAC method.

### 3.3.2 Analytical quantile mapping based on EVA


In the EVA-based quantile mapping, two POT-based extreme value distributions with different parameters
are matched. Being more specific, we want to construct a transformation $x \rightarrow y$ defined by requiring that
exceedance rates above $x$ and $y$, respectively, are equal for any $x$:

$$\lambda_x = \lambda_y.$$

This implies, according to (1), that

$$\lambda_{0x}[1 - \mathcal{G}_x(x - x_0)] = \lambda_{0y}[1 - \mathcal{G}_y(y - y_0)],$$

where $\mathcal{G}_x$ is the GPD distribution of the exceedances $x - x_0$ and $\lambda_{0x}$ the associated exceedance rate, and
$\mathcal{G}_y$ and $\lambda_{0y}$ are the similar entities for $y$.

To simplify, we let $\lambda_{0x} = \lambda_{0y}$ ( see Section 3.2) and therefore get

$$\mathcal{G}_x(x - x_0) = \mathcal{G}_y(y - y_0),$$

from which we obtain the transformation

$$y = y_0 + \mathcal{G}_y^{-1}\big(\mathcal{G}_x(x - x_0)\big). \text{ (3)}$$


For the delta-change approach (DC), the modelled GPD distribution functions for present-day and end-21[st]-
century conditions are quantile mapped and the transformation obtained this way is then applied to return
levels determined from present-day pseudo-observations $O_T$. Thus the corresponding projected $T$-year
return level is according to Eq. (3)

$$P_T = S_0 + \mathcal{G}_S^{-1}\big(\mathcal{G}_C(O_T - C_0)\big),$$

where $\mathcal{G}_C$ and $\mathcal{G}_S$ are the GPD cumulative distribution functions for the modelled present-day (control) and
end-21[st]-century (scenario) data, respectively, and $C_0$ and $S_0$ are the corresponding threshold values.

For the bias correction approach (BC), the present-day (control) and pseudo-observed GPD cumulative
distribution functions are quantile mapped to obtain the model bias, which is then applied, using eq. (3), to
modelled end-21[st]-century (scenario) return levels.

$$P_T = O_0 + \mathcal{G}_O^{-1}\big(\mathcal{G}_C(S_T - C_0)\big),$$

where $\mathcal{G}_O$ is the GPD cumulative distribution function for the observations and $O_0$ the corresponding
threshold.

### 3.3.3 Reference adjustment methods

The performance of the bias adjustment methods described above will be compared with the performance of two reference adjustment methods, which are defined below. This is a similar to what is practice when verifying predictions, where the performance of the prediction should be superior to the performance of reference predictions, such as persistence or climatology.

We choose two reference methods. One reference is to simply use, for a given model, the return level calculated from (pseudo-)observations as the projected return level (OBS),

$$P_T = O_T$$

Another reference is to use the raw scenario model output data without any adjustment (SCE):

$$P_T = S_T.$$

For an overview of methods, see Table 2

Table 2. Overview of methods used in the inter-comparison

| OBS | (Pseudo-)observations (Reference method) |
| --- | --- |
| SCE | Raw RCM scenario (Reference method) |
| FAC | Climate factor on return levels |
| DC | Quantile mapped delta-change based on EVA |
| BC | Quantile mapped bias correction based on EVA |

## 3.4 The inter-model cross-validation procedure in detail

The inter-model cross-validation goes in detail as follows: Each of the $N$ models are successively regarded as being pseudo-observations. The individual adjustment methods are calibrated on the present-day parts of the pseudo-observations and model return levels (present-day and end-21st-century), as appropriate depending on whether it is a bias correction or delta-change method. The calibration is done as described above. The adjustment methods are then applied to present-day observation and model data, again as appropriate, to obtain end-21st-century adjusted return levels. These are  then validated against the end-21st-century return level from pseudo-observations.

The basic validation metric will be the relative error of end-21[st]-century return levels for a given duration and return period $T$:

$$RE = |P_T - V_T|/V_T$$

i.e. the absolute difference between the projected return level $P_T$ obtained from using adjustment and the validation return level $V_T$ estimated from end-21[st]-century pseudo-observations, divided by the validation return level. This metric is calculated for every grid point and for every combination of model/pseudo-observations. Since we have $N = 19$ model simulations in the ensemble, we have $N \times (N - 1) = 342$

different combinations for validating each adjustment method and make statistics of the relative error. This
quantifies the average performance of the different methods.
User-end scenarios are often constructed as the median or mean from ensembles. We also tested this in
the inter-model cross-validation setup. The calibration is performed as before on each of the remaining
models and adjusted return levels for the end-21$^{st}$-century calculated. But then the median of these
adjusted future return levels is calculated and this is validated against the future pseudo-observations.
Note that this gives only $N = 19$ different combinations and therefore a less robust statistics compared to
above.

## 4   Results

### 4.1   Modelled return levels for present-day and end-21$^{st}$-century conditions

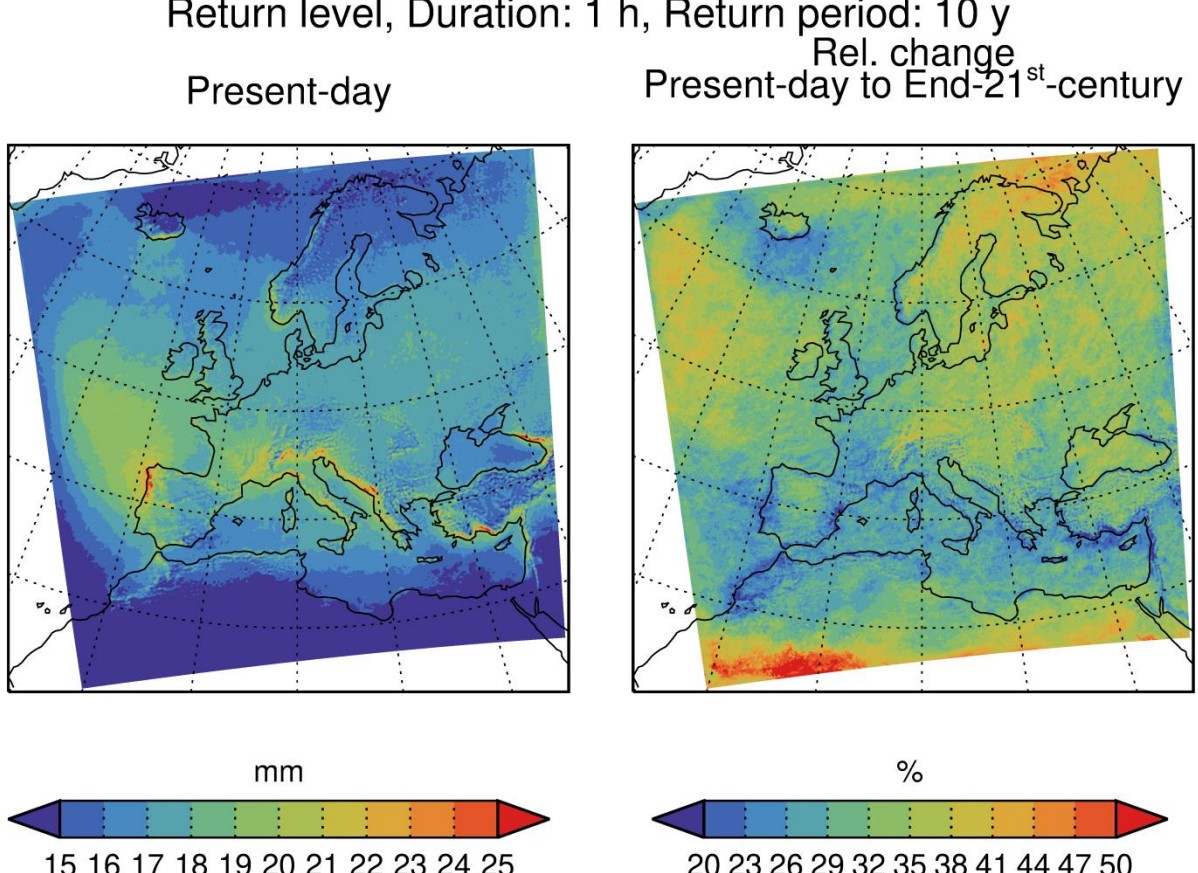

Figure 2. Geographical distribution of the 10 year-return level of precipitation intensity for 1 hour duration for present-day (left) and relative change from present-day to end-21$^{st}$-century (right). In each grid point, values are the median return level over all 19 model simulations.

Figure 2 displays the geographical distribution of the 10-year return level for precipitation intensity of 1 h
duration, calculated as the median return level over all 19 model simulations. The smallest return levels are
mainly found in the arid North African region and to some extent in the Norwegian Sea, while the largest
return levels are found in southern Europe and in the Atlantic northwest of the Iberian Peninsula.
Mountainous regions, such as the Alps and western Norway stand out as have higher return levels than
their surroundings. This supports that the models are not totally unrealistic in modelling extreme
precipitation.
There is a general increase in the range of 20-40% from present-day to end-21st-century climatic
conditions. The relative changes are geographically quite uniform across the area. For instance, no evident
difference between land and sea appears. Likewise do the mountainous regions not stand out from the
surroundings.

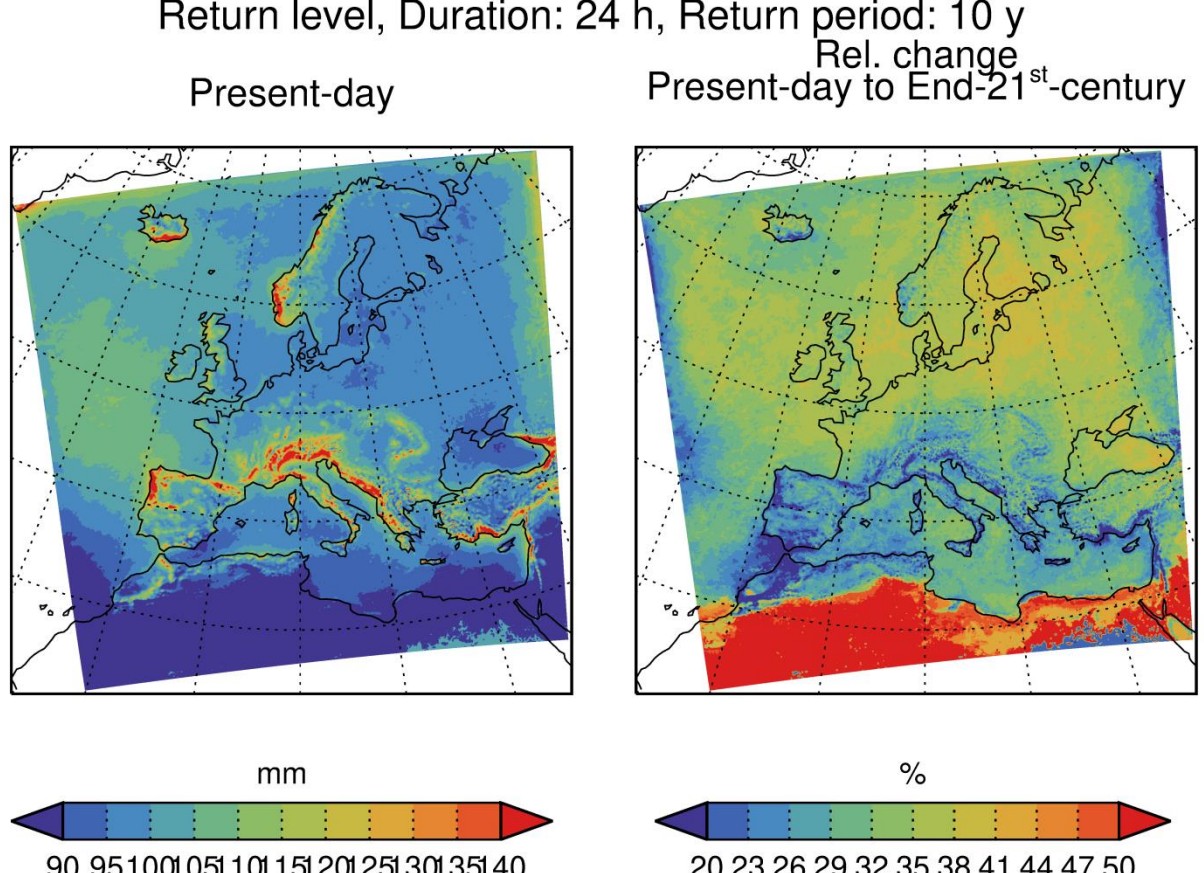

Figure 3. As Figure 2 but for 24 h duration

We also show in Figure 3 the median 10-year return level for 24 h duration. Again, the largest return levels
are found in southern Europe and northwest of the Iberian Peninsula. Also, the mountainous regions stand
out with higher return levels even more pronounced than for 1 h duration. The return levels generally
increase from present-day to end-21st-century conditions with around the same percentage as for 1 h
duration and also geographically homogeneous.

421    .

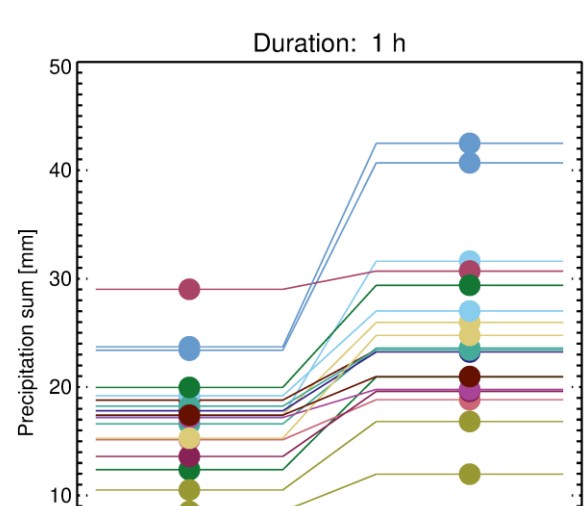
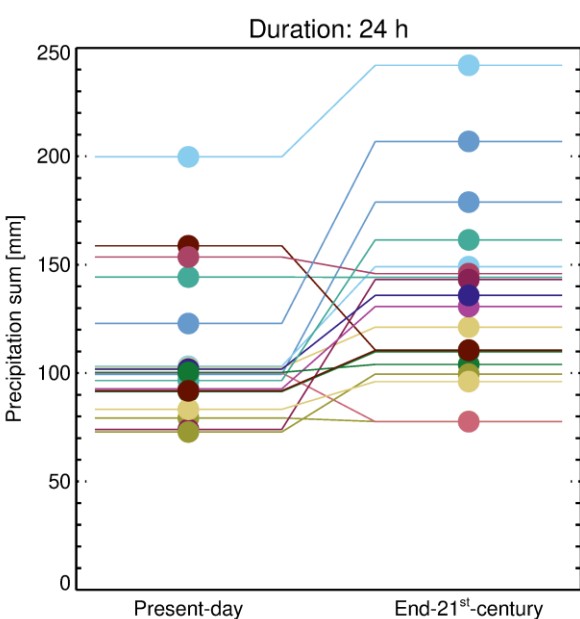

Figure 4. Modelled return levels at 50N/10E (northern Germany, marked with 'X' in Figure 1) for present and future for 10 y return
period and 1 h and 24 h durations. Different colours represent the 19 different GCM-RCM simulations listed in Table 1.

To get a more detailed impression of the data, Figure 4 shows return levels and their changes from present-
day to end-21$^{st}$-century for a grid point in Northern Germany for all 19 model simulations. For 1 h duration
(left panel) return values increase from present-day to end-21$^{st}$-century in all cases. For 24 h duration (right
panel) typically the return levels increase from present-day to end-21$^{st}$-century but with some exceptions.
This behaviour is common to all regions. For both durations, we also note the large spread in return levels
within the ensemble. The spread is much higher than the change between present and future for most
models; in other words: a poor signal to noise ratio. This is probably a combined effect of different climate
signals in different models and natural variability (Aalbers et al., 2018).

## 4.2  Inter-model cross-validation


In the following, we will present results using two different types of display. First, we will use spatial maps
of the median relative error, calculated from all combinations of model/pseudo-observations. Second, we
will, for each adjustment method and for each combination of model/pseudo-observations, calculate the
median relative error over each of the eight PRUDENCE sub-regions defined in Christensen and Christensen
(2007) and shown on Figure 1. For each region we will illustrate the distribution of the relative error across
all combinations of model/pseudo-observations by showing the median and the 5/95-percentiles of this
distribution.

### 4.2.1   Results for 1 h duration


Figure 5 shows the median, across all model/pseudo-observations combinations, the relative error for all
five methods for 1 h duration and 10 y return period.

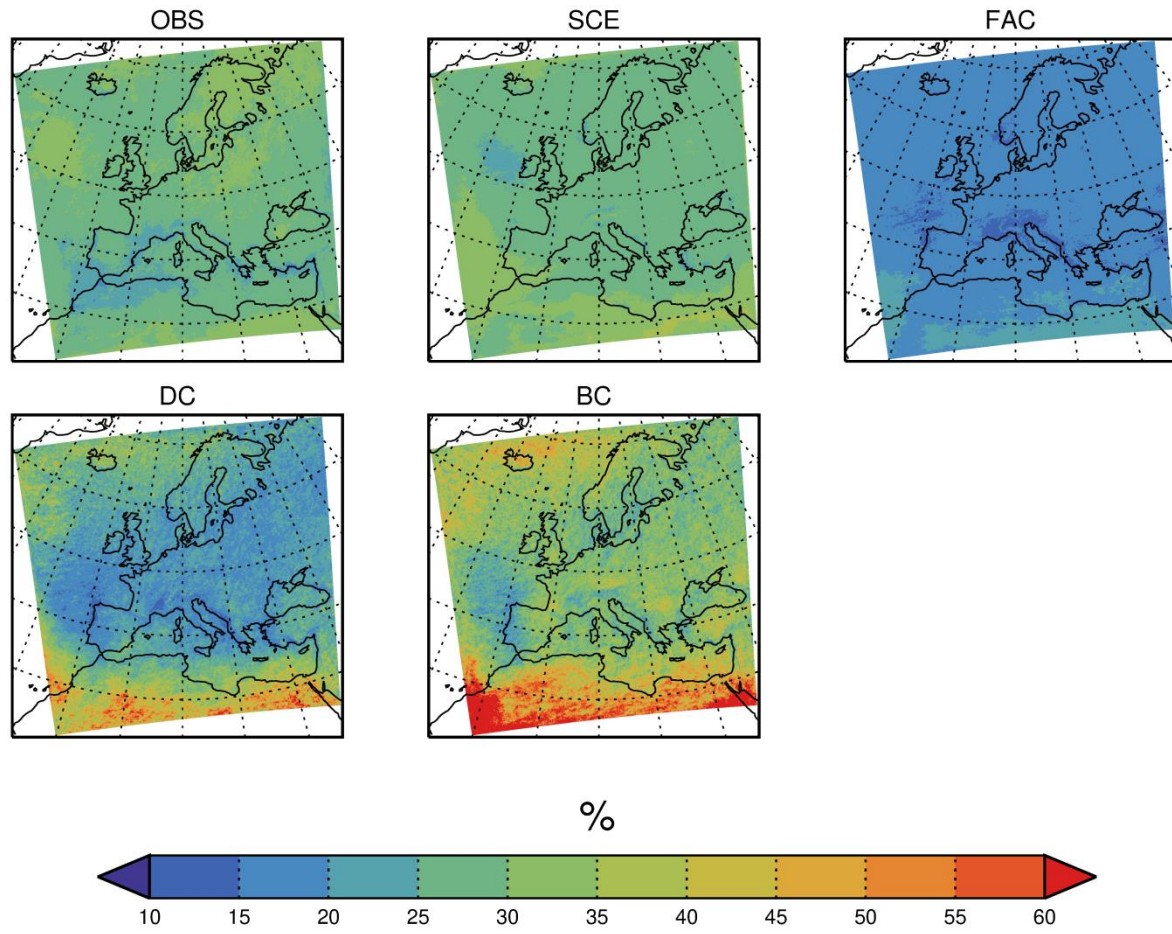

Figure 5. Geographical distribution of the relative error of end-21[st]-century 10 year return level for 1 h duration precipitation
intensity from the inter-model cross-validation. Colours show the median of the relative error calculated over all model/pseudo-
observations combinations. Panels are for the different adjustment methods.

First we look at the reference methods. Relative errors from the OBS method are in the range of 20-40%.
Lowest values are found in the Mediterranean, western France and the Atlantic west of the Mediterranean;
highest values in the Atlantic west of Ireland and in Scandinavia. The SCE method has errors in the interval
25-45%, lowest values in the Atlantic west of Ireland; largest values over parts of the Atlantic and northern
Africa. The two reference methods give rather similar results, but the OBS method slightly outperforms SCE
in the south, while the opposite is true in the north.

The relative error of FAC is below 20% in most places. It is everywhere smaller than the relative error of the
reference methods OBS and SCE. The DC method has a relative error comparable to (e.g. Western France,
Western Iberia and Eastern Atlantic) or larger than (in particular in Northern Africa) that of FAC. That said,
the concept of relative error should be used with care in an arid region, such as Northern Africa. But from
this result, it is not justified to use the more complicated DC, in favour of the simpler FAC. Finally, the
relative error of BC is everywhere above both DC and FAC, indicating the poorest performance of all
methods considered.

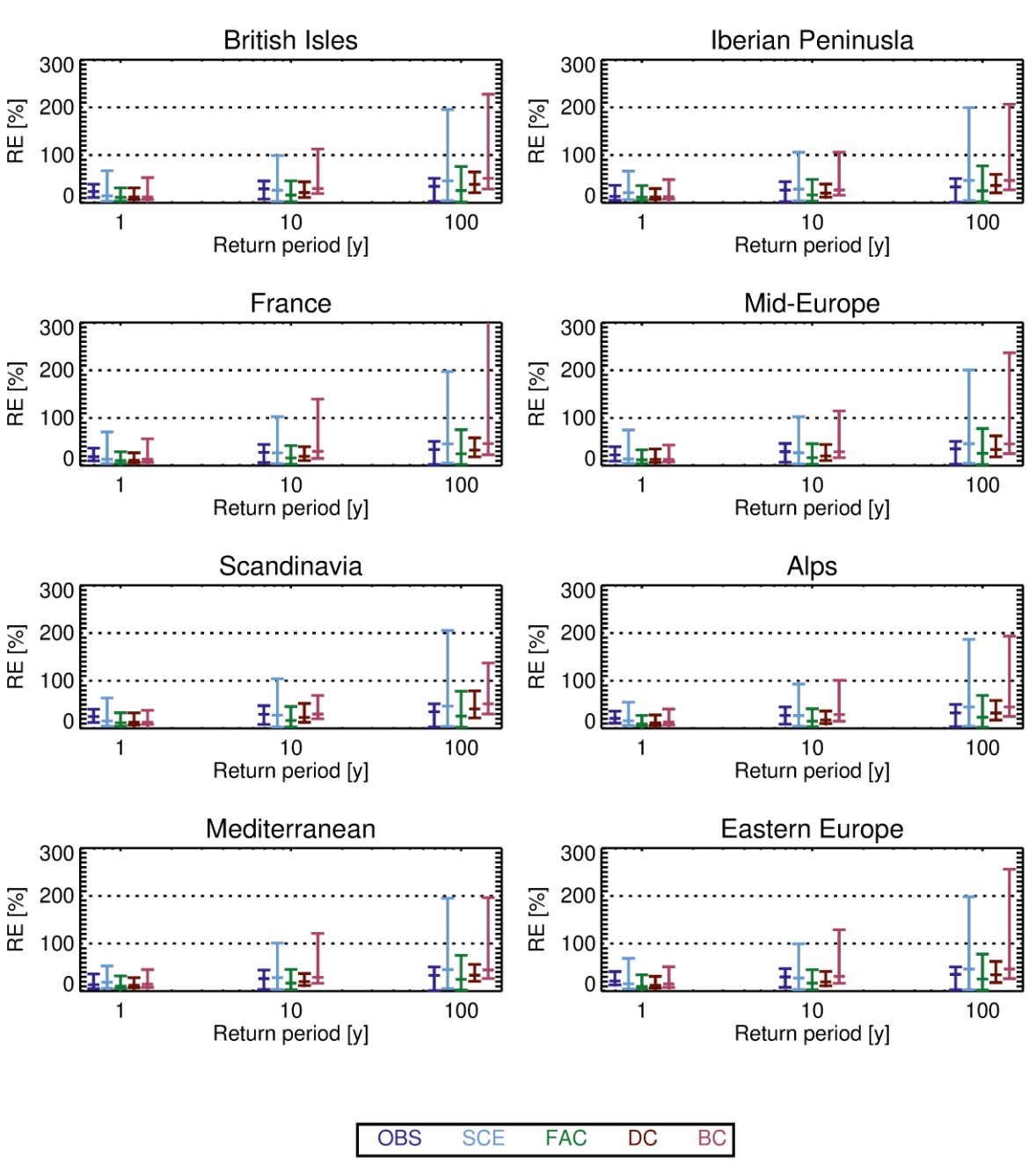


Figure 6. Statistical distribution (median and 5[th]/95[th]  percentile) of the relative error of the inter-model cross-validation for 1 hour
duration for 1 y, 10 y and 100 y return periods.  Panels represent PRUDENCE sub-regions shown in Figure 1.  Each colour represents
a adjustment method (see Table 2).

The statistical distribution of the relative error is shown in Figure 6 for the eight PRUDENCE sub-regions
(see Figure 1). We first note that the distribution of relative error is shifted towards higher values for larger
return periods, as expected. Next, we note that the two reference methods, OBS and SCE, behave
differently. SCE generally has a little larger median relative error, but the 95[th] percentile is much larger for
SCE than for OBS, in particular for large return periods. Thus, OBS overall performs better than SCE,
meaning that using present-day pseudo-observations to estimate projected end-21[st]-century return levels
yields better relative error than using raw modelled scenario data.

The FAC method generally has the best overall performance, both in terms of median and 95[th] percentile of
the relative error. The DC method has a slightly poorer performance than FAC, both in terms of the median
and the 95[th] percentile of the relative error. Finally, BC has poorer performance than DC, when comparing
the median of the relative error and in particular for the 95[th] percentile.

In summary, for 1 h duration, the method with the best performance is using a climate factor on the return
levels (FAC). This method outperforms both reference methods and the more sophisticated methods based
on quantile mapping, DC and BC, the latter having the poorest overall performance of them all. Note that
DC is comparing GPDs from the same model, whereas BC is comparing GPDs from different models. If the
difference, in terms of GPD parameters, between two models in the present-day climate is typically larger
than the difference between the same model in present-day and end-21st-century climate, it can explain
the different results.


**4.2.2   Results for 24 h duration**

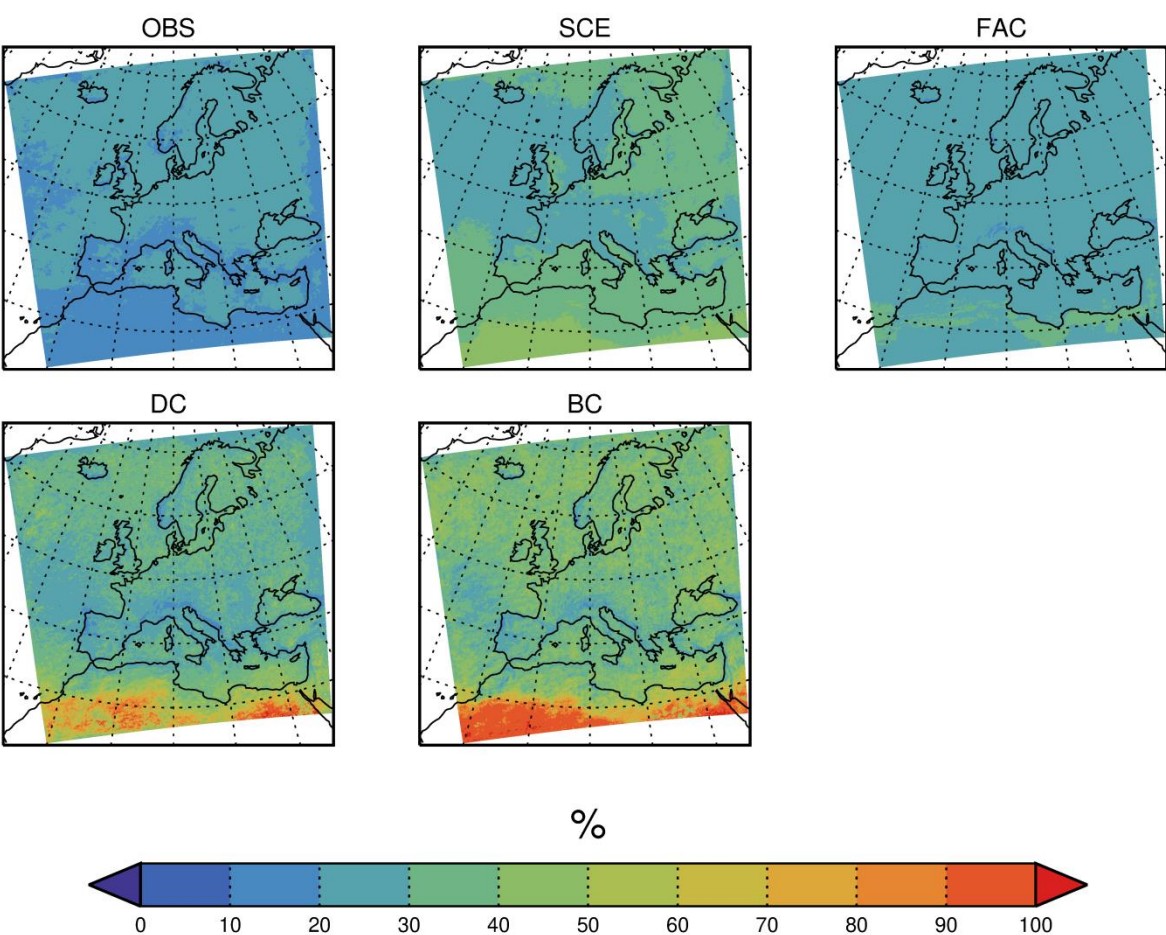

Relative error, Duration: 24 h, Return period: 10 y

OBS SCE FAC

DC BC

%

0  10  20  30  40  50  60  70  80  90  100

Figure 7. As Figure 5 but for 24 h duration.

For 24 h duration (see Figure 7 ), OBS has the lowest median relative error (less than 30%) in most regions
of all the adjustment methods, while SCE has higher relative error in the interval 30-60% approximately,
with the highest values in North Africa. FAC has relative errors in-between those of OBS and SCE. Of the
quantile mapping methods, DC has relative errors in the interval 20-80% approximately, larger than FAC in
most places, and finally BC has, as for 1 h duration, the largest median relative errors of all the methods.

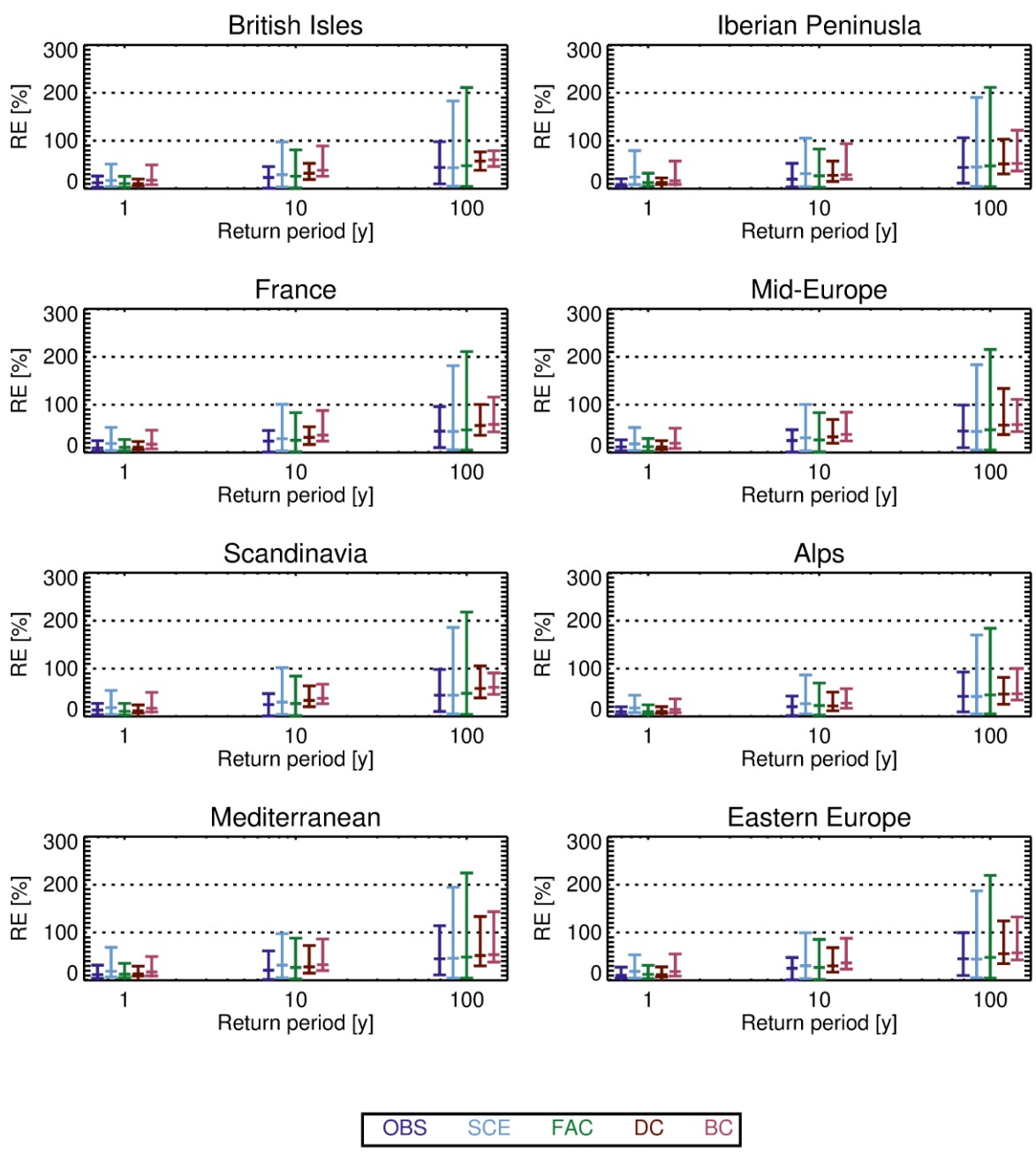

Figure 8. As Figure 6 but for 24 h duration

As for the 1 h duration, we also compare the entire statistical distribution of the relative error of the different adjustment methods for all three return periods (Figure 8), and again, both median and 95[th] percentile of the relative error increases for larger return periods, as expected. Further, OBS seems, surprisingly, to have a small median relative error and the smallest 95[th] percentile of all methods considered for all sub-regions. SCE has a median not too different from that of OBS, but the 95[th] percentile

is much larger. Similar characteristics hold for FAC. The quantile mapping methods DC and BC have slightly
larger median values, but the 95[th] percentile is smaller than for FAC. All these characteristics hold for all
sub-regions.

### 4.2.3 Ensemble median

Also inter-model cross-validation of pseudo-observations against model ensemble median, as described in
Section 3.4, was carried out. For duration 1 h, distribution of the relative error is shown in Figure 9. By
comparing with Figure 6, the distribution of the relative error does not change much overall.  However, for
many of the sub-regions considered and for the longer return periods, the FAC and BC have a smaller 95[th]
percentile for cross-validation against model ensemble means, than against individual models.

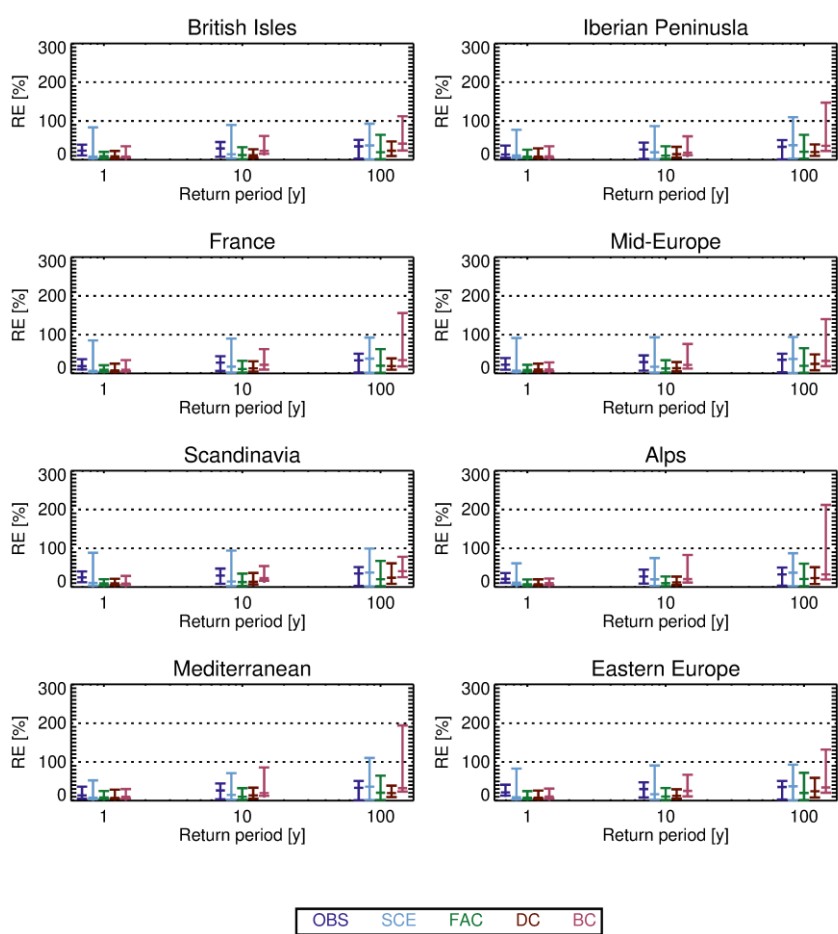

Figure 9. As Figure 6 but for inter-model cross-validation against ensemble medians.

Also for 24 h duration the distribution of the relative errors does not change much when shifting to
validation against ensemble median (not shown).

## 4.3 Further analysis on conditions for skill


To get further insight into the difference in performance between hourly and daily precipitation, we
consider for a given return period the relationship between the bias factor for present-day $B_{P,T} = \frac{C_T}{O_T}$ and
end-21[st]-century $B_{F,T} = \frac{S_T}{V_T}$ for all model/pseudo-observations combinations (see Figure 10).

## Bias factor of return level
### Region: Mid-Europe, Return period: 10 y

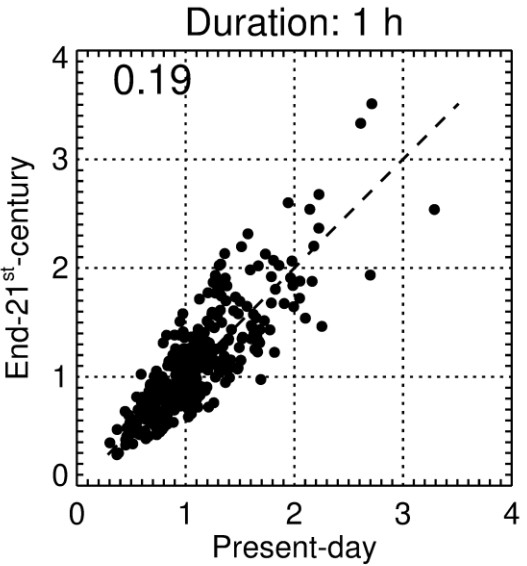
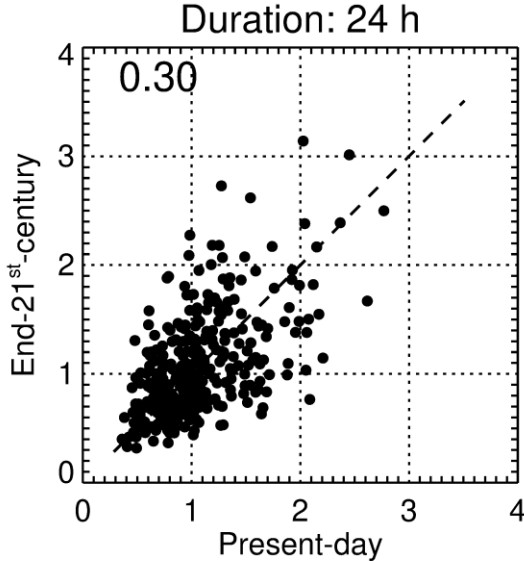

Figure 10. Relationship between present-day and end-21[st]-century bias factors of 10-year return levels for Mid-Europe sub-region
for all pseudo-observation/model combinations. Left panel: 1 h duration and right panel: 24 h duration. Numbers in upper left
corners are the **R** indices. See text for details.

In this figure, the relationship between present-day and end-21[st]-century bias factors appears more
pronounced for 1 h duration than for 24 h duration. That said, it must be borne in mind that if the point
$(x, y)$ is in the plot, so is the point $(1/y, 1/x)$, and this implies an inherent tendency to a fan-like spread of
points from $(0,0)$, as seen on both plots.

To quantify the strength of the above relationship, we define an index:
$$R = \langle \frac{|B_F - B_P|}{(B_F + B_P)/2} \rangle,$$
where $\langle \cdot \rangle$ means averaging over combinations of model/pseudo-observations. This index is an extension of
the index introduced by Maurer et al. (2013). It is the ensemble average of the relative absolute difference
between the present-day and future bias. A value of $R = 0$ means these biases are equal, i.e. perfect
stationarity; and the smaller the value of $R$, the closer to stationarity (in an ensemble sense).

Values of $R$ are given in the upper left corner of each panel of Figure 10 and they also support the partial
relationships described above, and a stronger one for hourly duration. These relations are important since
they could explain the generally good performance of the FAC method seen in the previous section.
Suppose that $B_{P,T} = B_{F,T}$ , then
$P_T = \frac{S_T}{C_T} O_T = S_T \frac{O_T}{C_T} = S_T B_P = S_T B_F = S_T \frac{V_T}{S_T} = V_T$
and the FAC method will therefore adjust perfectly.
We also note that daily data, due to the summation, would have less erratic behaviour than hourly and
therefore we would expect any relationship to be less masked by noise for daily data than for hourly data
from purely statistical grounds. Therefore, any explanation to why it is opposite should probably be found
in physics or details of modelling. We will discuss this further in Section 5.3.
# 5   Discussion

## 5.1   Relation with other studies

The study by Räty et al. (2014) touches upon related issues to ours. However, our study includes smaller
temporal scales (hourly and daily) and higher return periods (up to 100 years vs. the 99.9$^{th}$ percentile of
daily precipitation corresponding to a return period of around 3 years). Nevertheless, the two studies agree
in their main conclusion; namely that applying a bias adjustment seems to offer an additional level of
realism to the processed data series, including in the climate projections, as compared to using unadjusted
model results. The two studies both support, in agreement with our study, the somewhat surprising
conclusion that using present-day (pseudo-)observations as the scenario gives a skill comparable to that of
the bias adjustment methods.
Kallache et al. (2011) proposed a correction method for extremes, CDF-t, and obtained good validation
result with calibration/validation split of historical data from Southern France. The CDF-t method was
applied by Laflamme et al. (2016) on daily New England data and concludes that "downscaled results are
highly dependent on RCM and GCM model choice".

## 5.2   Convection in RCMs
The grid spacing of present state-of-the-art RCMs available in large ensembles, such as CORDEX, is around
10 km, and at this resolution it is necessary to describe convection through parameterizations. This is
obviously an important deficit for our purpose, since this could represent a systematic bias in all our
simulations and therefore violate our underlying assumptions that the individual model simulations and the
real-world observations behave similarly in a physical sense. Thus, we do not promote naively applying the
presented adjustment methods to hourly data from these models. Instead, the present work should be
seen as a statistical exercise and the methods can in the future be applied to convection permitting model
simulations that better represent the convective process. The results from the present work would apply
equally to that case.
With the advent of convective-permitting models, a more realistic modelling of convective precipitation
events is within reach and a change in the characteristics of such events is seen (Kendon et al., 2017;
Lenderink et al., 2019; Prein et al., 2015). This next generation of convection-permitting RCMs with a grid
spacing of a few km allows a much better representation of the diurnal cycle and convective systems as a
whole (Prein et al., 2015). With that in mind, we foresee redoing the analysis when a suitable ensemble of
convective-permitting RCM simulations becomes available.

## 5.3 Stationarity of bias

The success of applying bias adjustment to climate model simulations is linked to the biases being
stationary, i.e. present and future biases being more or less identical. In Section 4.3 we showed (in Figure
10) that this was the case for 1 h duration and less so for 24 h duration in our pseudo-reality setting. Such a
relationship is an example of an emergent constraint (Collins et al., 2012). This is a model-based concept,
originally introduced to explain that models which have a too warm (cold) present-day climate tend to have
a relatively warmer (colder) future climate. The reason for this is that it is the same underlying physics
which generates the present-day and future temperatures (Christensen and Boberg, 2012).
We suggest that our observed emergent constraints could be explained in a similar manner; namely as a
result of the Clausius-Clapeyron relation linking atmospheric temperature changes to changes in its
humidity content and thereby precipitation changes. The change prescribed by the Clausius-Clapeyron
equation is usually termed the thermodynamic contribution. In addition to this, there is a dynamic
contribution and this may explain the differences between the hourly and daily relation seen in Figure 10.
The rationale is that hourly extremes are entirely due to convective precipitation events with almost no
dynamic contribution (Lenderink et al., 2019), while daily extremes are a mixture of convective events and
large-scale strong precipitation, of which the latter has a more significant dynamic contribution (Pfahl et al.,
2017), causing the less marked emergent constraint for the daily time scale. This interpretation is also
supported in Figure 4, in which daily precipitation sees some 'crossovers' (future return level smaller than
present), whereas hourly precipitation does not have any crossovers.

## 5.4 The spatial scale

In the definition of model bias it is tacitly assumed that the observational dataset has the same spatial
resolution as the model data. In practice, however, it is rarely possible to separate the bias from a spatial
scale mismatch. For instance, if we compare modelled precipitation, which represents averages over a grid
box, with rain gauge data, which represent a point, there can be a quite substantial mismatch for extreme
events (Eggert et al., 2015; Haylock et al., 2008). Therefore, if the bias is adjusted towards such point
values, it may lead to further complications (Maraun, 2013).
Sometimes though, it is desirable to include the scale mismatch in the bias adjustment. Many impact
models, e.g. hydrological models, are tuned to perform well with local observational data as input. This
presents an additional challenge if this impact model is to be driven by climate model data for climate
change studies, since the climate model will have biases in its climate characteristics (mean, variability, etc.)
compared to those of the observed data. Applying the adjustment step, the hydrological model can rely on
its calibration to observed conditions (Haerter et al., 2015; Refsgaard et al., 2014).

## 5.5 Adjustment methods not included in the study

Only the basic adjustment methods have been included in our study. The simple climate factor approach
has been applied in numerous hydrological applications (DeGaetano and Castellano, 2017; Sunyer et al.,
2015) and others. We also wanted to test quantile mapping approaches, which in extreme value theory
takes the form of a parametric transfer function. This we have applied in two flavours in the spirit of (Räty
et al. (2014). Finally, we wanted to benchmark against the 'canonical' benchmark methods: observations
and raw model output.
There is a myriad of more specialised methods, each tailored to account for a particular deficit of the
simpler methods. First, there is the issue whether it for precipitation is more reasonable to map relative
quantile changes rather than absolute ones (Cannon et al., 2015). It has also been argued that a bias
correction method should preserve long-term trends, i.e. the 'climate signal' and only adjust the shorter
time scales, as extensively discussed in (Cannon et al., 2015). Then multivariate methods have been argued
for and applied in order to preserve relationships between variables (Cannon, 2018), and nested methods
to account for different biases for different time scales (Mehrotra et al., 2018). Also methods to correct for
systematic displacement of variable features in complex terrain have been suggested and applied (Maraun
and Widmann, 2015). Finally, Li et al. (2018)  adjusts stratiform and convective precipitation separately
instead of adjusting the total precipitation. In this way, any future change in the ratio between the two
types of precipitation is accounted for.
It could be interesting to examine the above methods in future studies, though we acknowledge it would
be a quite extensive work. We can at present only guess about the outcome of such work but the more
refined methods may not perform too well in the inter-model cross-validation setting. The reason for this
suspicion is that these methods, while being more elaborate, in most cases also have more parameters to
be estimated, implying a higher risk of overfitting. An argument in favour of this is that the present study
shows that the more elaborate quantile mapping methods DC og BC do not outperform the simpler FAC
method.

# 6   Conclusions

Based on hourly precipitation data from a 19-member ensemble of climate simulations we have
investigated the benefit of bias adjusting extreme precipitation return levels on hourly and daily time scales
and evaluated the different methods. This is done in a pseudo-reality setting, where one model simulation
in turn from the ensemble plays the role of observations extending into the future. The return levels
obtained from each of the remaining model simulations are then adjusted in the present-day period, using
different adjustment methods. Then the same adjustment methods are applied to end-21$^{st}$-century model
data to obtain projected return levels, which are then compared with the corresponding pseudo-realistic
future return levels.
The main result of this inter-comparison is that applying bias adjustment methods improves projected
extreme precipitation return levels, compared to using the un-adjusted model runs. Can an overall superior
adjustment methodology be appointed? For hourly duration, the method to recommend (having the
smallest relative error) is the simple climate factor approach FAC, which is better in terms of the relative
error than the more complicated analytical quantile mapping methods based on EVA, DC and, in particular,
BC. For daily duration, the OBS method performs surprisingly well, having the smallest 95th percentile of
the relative error. Furthermore, the quantile mapping methods perform better than FAC, with DC having
the smallest relative error. These conclusions hold regardless of the sub-region considered. We also cross-
validated against model ensemble means; this gave in general similar results without significant changes in
the distribution of the relative error.
Finally, we registered emergent constraints between present-day and end-21$^{st}$-century biases. This was
more pronounced for hourly than for daily time scales. This could be caused by hourly precipitation being
more directly linked to the Clausius-Clapeyron response, but this requires more clarification in future work.
*Data availability.* The hourly EURO-CORDEX precipitation data are not part of the standard suite of CORDEX
and are therefore not produced nor shared by all modelling groups. The data used in this study may be
obtained upon request from each modelling group. The IDL code used in the analysis can be obtained from
TS.
*Author contribution.* TS and PT designed the analysis with contribution from other co-authors and
programmed the analysis software. PB, FB, OBC and PT prepared the data. TS prepared the manuscript with
contributions from PT, PB, FB, OBC, BC, JHC, CS, and MSM.
*Competing interests.* The authors declare that they have no conflict of interest.

*Acknowledgements.* The work was supported by the European Commission through the Horizon 2020
Programme for Research and Innovation under the EUCP project (Grant Agreement 776613). Part of the
funding was provided by the Danish State through the Danish Climate Atlas. PB was funded by the project
AQUACLEW, which is part of ERA4CS, an ERA-NET initiated by JPI Climate, and funded by FORMAS (SE), DLR
(DE), BMWFW (AT), IFD (DK), MINECO (ES), ANR (FR) with co-funding by the European Commission (Grant
Agreement 690462). Some of the simulations were performed in the COPERNICUS C3S project C3S_34b
(PRINCIPLES). We acknowledge the World Climate Research Programme's Working Group on Regional
Climate, and the Working Group on Coupled Modelling, former coordinating body of CORDEX and
responsible panel for CMIP5. We thank the climate modelling groups (listed in Table 1 of this paper) for
producing and making their model output available. We also acknowledge the Earth System Grid
Federation infrastructure, an international effort led by the U.S. Department of Energy's Program for
Climate Model Diagnosis and Intercomparison, the European Network for Earth System Modelling and
other partners in the Global Organisation for Earth System Science Portals (GO-ESSP). It is appreciated that
Geert Lenderink, KNMI, Claas Teichmann, GERICS and Heimo Truhetz, University of Graz made model data
of hourly precipitation available for analysis. We appreciate constructive comments from referee Jorn van
de Velde, from two anonymous referees, and from T. Kelder, R. L. Wilby, T. Marjoribanks, and L. Slater.

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
