# Peer review of "Identifying robust bias adjustment methods for extreme precipitation in"

_Hydrology and Earth System Sciences, 2020_

## Referee Comment (RC1) · Anonymous Referee #1 · 21 Jul 2020

This is an interesting contribution involving a lot of work. I have a few general issues that the authors should address in their revisions, followed by some specific comments.

Firstly - there needs to be a better discussion about the possible problems in using the psuedo-reality setting for assessment of precipitation extremes. Most models have a tendency to increase the probability of occurrence of rainfall, thereby increasing the size of the sample that could potentially constitute extremes. The authors have avoided this issue to some extent by performing a psuedo-reality assessment. I believe some discussion should be included as this could create difficulties in taking the findings from here to real applications.

Secondly, the paper is coming across as a bit of a report (and I sympathise with the authors as they do have a lot of information to present). Perhaps a more creative

discussion for differences in mountaineous areas versus not, coastal areas versus not, and daily durations versus hourly would be useful. I note the spatial resolution is 11km. Daily extremes should be simulated better at this resolution. Also, no mention is made of the causative GCMs that are interpolated using the RCMs. There are different extent of biases in these. Some discussion should be included on this as well.

Thirdly, the authors have missed with publications on this topic by Jingwan Li. Relevant papers are: Li, J., et al. (2017). "A comparison of methods for estimating climate change impact on design rainfall using a high-resolution RCM." Journal of Hydrology 547: 413-427. Li, J., et al. (2017). "A comparison of methods to estimate future subdaily design rainfall." Advances in Water Resources 110: 215-227. Li, J., et al. (2018). "Addressing the mischaracterization of extreme rainfall in regional climate model simulations – A synoptic pattern based bias correction approach." Journal of Hydrology 556: 901-912. Li, J., et al. (2018). "Can Regional Climate Modeling Capture the Observed Changes in Spatial Organization of Extreme Storms at Higher Temperatures?" Geophysical Research Letters 45(9): 4475-4484.

I am a co-author on these papers hence have a conflict here. But I think these are very relevant to what the authors are attempting to do here, as she used an even finer resolution RCM with a high density of observed gauges at the same time resolution (hourly). The bias correction approach she adopted acknowledged the bias in simulating convection within the RCMs as well as the quantile bias convective and non-convective rainfall were exhibiting.

Now to the specific comments:

line 142 - missing section marker

line 225 - there is another way to create the partial series sample. It is to acknowledge that there may be a bias in the proportion of events that are say convective. If this proportion is biased, one is forming a biased sample effectively by selecting the series the way adopted here. This issue is the focus of Li, J., et al. (2018). "Addressing

the mischaracterization of extreme rainfall in regional climate model simulations – A synoptic pattern based bias correction approach." Journal of Hydrology 556: 901-912.

line497 - If the proportion of convective extreme events increases in the future (as it is expected to) then ignoring any bias in the representation of convection as discussed above, will create a non-stationary bias. This can be addressed though using the above mentioned approach.

---

## Author Comment (AC1) · 14 Aug 2020

Authors' response to referee #1

* We will start by thanking the referee for a fair and thorough review. We will comment (marked with * . . . *) on each review items below. *

This is an interesting contribution involving a lot of work. I have a few general issues that the authors should address in their revisions, followed by some specific comments. Firstly - there needs to be a better discussion about the possible problems in using the pseudo-reality setting for assessment of precipitation extremes. Most models have a tendency to increase the probability of occurrence of rainfall, thereby increasing the size of the sample that could potentially constitute extremes. The authors have avoided

this issue to some extent by performing a pseudo-reality assessment. I believe some discussion should be included as this could create difficulties in taking the findings from here to real applications.

* We will clarify the introduction by underlining that bias adjustment methods can be validated by two main methodologies: 1) training/validation split of observations (Themeßl et al. 2011; Li et al. 2017) and 2) inter-model cross-validation, which we use. Methodology 1) has the advantage of actually comparing the bias-adjusted model date with observations, which methodology 2) does not. This may be important, since models may have precipitation characteristics, e.g. the division between stratiform and convective precipitation, different from observations. Methodology 2), on the other hand, also has advantages. Firstly, the models are tested in in future climatic conditions, different from present-day conditions on which they are trained. Also, model and pseudo-reality have the same spatial scale, thus avoiding any interfering with scale issues present in methodology 1) where pointwise observations are compared with area-averaged model data.

We recognize that models do have a tendency to increased probability of rainfall. This is, however, in the form of continuous drizzling and therefore would not affect our extreme value analysis. *

Secondly, the paper is coming across as a bit of a report (and I sympathise with the authors as they do have a lot of information to present). Perhaps a more creative C1 discussion for differences in mountaineous areas versus not, coastal areas versus not,

* Our adoptation of the 'PRUDENCE regions' was motivated by the desire to illustrate results for different meteorological regimes. This we will formulate more explicitly. We will also consider using Figs. 5 and 7 (with a revised color scale) for relating the results to storm track, orography and coastlines. *

and daily durations versus hourly would be useful. I note the spatial resolution is 11km. Daily extremes should be simulated better at this resolution.

\* We will extend the text in the results in Section 4.2. on daily vs. hourly. B.t.w., the issue is dealt with in more principal terms in Section 4.3. \*

Also, no mention is made of the causative GCMs that are interpolated using the RCMs. There are different extent of biases in these. Some discussion should be included on this as well.

\* We will include this discussion, based on the available literature on GCMs and their ability to reproduce large-scale features (storm tracks, blocking frequency and, duration) and RCMs and their ability to reproduce local details of precipitation due to e.g. orography. A priori, we regard the effect from the RCM to be important for extremes.

We note that only few pairs of pseudo-reality and model use the same GCM model and ensemble member. Counting in Tab. 1 shows that 40 pairs out of 342 are driven by the same GCM ensemble member, i.e. 11% of the pairs analyzed. A specific analysis of the relation between the two kinds of model pairs would be very interesting, but it is not easily included into the current study. \*

Thirdly, the authors have missed with publications on this topic by Jingwan Li. Relevant papers are: Li, J., et al. (2017). "A comparison of methods for estimating climate change impact on design rainfall using a high-resolution RCM." Journal of Hydrology 547: 413-427. Li, J., et al. (2017). "A comparison of methods to estimate future subdaily design rainfall." Advances in Water Resources 110: 215-227. Li, J., et al. (2018). "Addressing the mischaracterization of extreme rainfall in regional climate model simulations – A synoptic pattern based bias correction approach." Journal of Hydrology 556: 901-912. Li, J., et al. (2018). "Can Regional Climate Modeling Capture the Observed Changes in Spatial Organization of Extreme Storms at Higher Temperatures?" Geophysical Research Letters 45(9): 4475-4484. I am a co-author on these papers hence have a conflict here. But I think these are very relevant to what the authors are attempting to do here, as she used an even finer resolution RCM with a high density of observed gauges at the same time resolution (hourly). The bias correction approach

she adopted acknowledged the bias in simulating convection within the RCMs as well as the quantile bias convective and non-convective rainfall were exhibiting.

* We were not aware of these papers. We foresee that the two papers "A comparison . . ." will be referenced in Section 5.1 and contribute to the discussion there.

Our manuscript evaluates basic adjustment methods only. We know that there is a myriad of special-designed adjust methods, including the one described in the paper "Addressing the mischaracterization . . . ".

The paper "Can Regional Climate Modeling Capture . . ." about the spatial extent of extreme precipitation events is not within the scope of our manuscript. *

Now to the specific comments:

line 142 - missing section marker

* Thanks, will be fixed. *

line 225 - there is another way to create the partial series sample. It is to acknowledge that there may be a bias in the proportion of events that are say convective. If this proportion is biased, one is forming a biased sample effectively by selecting the series the way adopted here. This issue is the focus of Li, J., et al. (2018). "Addressing the mischaracterization of extreme rainfall in regional climate model simulations – A synoptic pattern based bias correction approach." Journal of Hydrology 556: 901-912.

* In the manuscript we evaluate the standard POT procedure, as it is implemented by numerous hydrological authorities. The work described in the suggested paper is not within our scope (see also above). *

line497 - If the proportion of convective extreme events increases in the future (as it is expected to) then ignoring any bias in the representation of convection as discussed above, will create a non-stationary bias. This can be addressed though using the above mentioned approach.

\* The aim of our work is to evaluate the simple bias adjustment methods for extremes, as also explained above. More sophisticated methods are not included in this study, but the suggested paper can go into the discussion on future work. \*

---

## Referee Comment (RC2) · Jorn Van de Velde (Referee) · 14 Sep 2020

Jorn Van de Velde (Referee)

jorn.vandevelde@ugent.be

General comments In their contribution, Schmith et al. (2020) discuss the robustness of different bias-adjusting methods for (sub)daily rainfall extremes. This yields interesting results and strong links with the context of convection-permitting models and emergent constraints. Yet, there are some aspects about whom I'd like a deeper discussion.

The first aspect is the practical use of this study. This is foremost linked with the choice of bias-adjusting methods. Although the use of return periods is perfectly justified from a hydrological point of view, I've seen few studies that actually use bias adjustment directly on the return periods. As such, I'd like to see a larger discussion on the choice of bias-adjusting methods. Given a well-justified choice, I understand the use of these

simple methods, yet I'd like to see more discussion on how this relates with more complicated, but related bias-adjustment methods, such as e.g. CDF-t (Michelangeli et al., 2009), standard QM, QDM (Cannon et al., 2015), … Would it be possible to discuss possible consequences for the use of these methods for the adjustment of subdaily precipitation extremes? This could fit in the second paragraph of Section 5.1, which seems rather limited and abrupt at this point. A last point related to the practical use is that I missed a more thorough explanation of why the observations perform well, why this version of quantile mapping performs poorly. Although this is discussed slightly in Section 4.3, I wonder if more details or, if possible, practical guidelines could be given in the discussion.

A second aspect is that some concepts in the Introduction seem to be accepted as-is, whereas they could deserve a deeper discussion. A first example of this is the discussion of stationarity in the introduction. The references are limited in time, whereas more recent papers expanded this subject, such as Kerkhoff et al. (2014) and Van Schaey-broeck and Vannitsem (2016) on the type of bias relationship and Chen et al. (2015), Velázquez et al. (2015), Wang et al. (2018) and Hui et al. (2019), who discussed the uncertainty introduced by bias nonstationarity. As the stationarity of the bias is an important part of the discussion, I think the paper could benefit from these perspectives. A second, smaller example is the use of a delta change based method. While the method isn't completely discredited, there has been some discussion whether it's use for climate change is not too dependent on the assumption that the temporal structure of the time series will not change from present to future (e.g. Johnson and Sharma (2011), Kerkhoff et al. (2014)). It would thus be interesting to read a deeper discussion on the limitations of the methods.

Specific comments

L. 37: 'quantile-mapping' is used here, whereas in the remainder of the abstract (and the paper) 'quantile-matching' is used. I'd suggest to edit this for coherence, but to also use 'quantile mapping' throughout the paper, as it has been the most used term for this

type of bias adjustment during the last few years.

L. 75-82: this paragraph is very scarce on references. Although some of the necessary references are given in the discussion, I think it would be good to also have the reference to the papers about CPMs in this paragraph.

L. 84-91: The terminology in this paragraph could be reconsidered. Although it is debatable whether or not to consider delta change as a bias adjustment approach (the latest textbook, Maraun and Widmann (2018), is on the edge), it feels very strange to read 'bias correction' as a subset of 'bias adjustment' approaches. The use of 'bias adjustment' as a replacement of 'bias correction' has been rising during the last few years, as it is clearer that the methods are statistical and cannot correct all climate model biases. Thus, I would withhold from the use of 'bias correction'. Better terminology seems MOS, with delta change and bias adjustment as possible subcategories, or bias adjustment with delta change and bias adjustment s.s., although the exact choice is personal.

L. 253- 286: Although the method described here is indeed based on the same principles as XCDF-t as used by Kallache et al. (2011) and Laflamme et al. (2016), it's not entirely clear how the new method is created by adapting the former. I think the link between both methods should be more detailed, so users can retrace it more easily and infer the strengths and limitations. Especially as it is specifically mentioned that the method 'will be adapted to our needs below', the adaptation seems rather limited.

L. 448-453: the explanation of the use of the index by Maurer et al. (2013) should be expanded. Firstly, it's unclear to me where the terminology 'measure of relative spread' is derived from, as it is not named as such in the original paper. Secondly, the interpretation of the R-values is not discussed, although this is quite important: values < 1 indicate that the difference in biases is smaller than the mean bias of both periods, whereas values >1 indicate that the difference in biases is larger, which could have a potentially large impact. As both values are quite far < 1, the bias seems quite

stationary, but in your discussion you state that the 24h duration is 'less stationary'. Without giving this numerical explanation, this statement is hard to interpret correctly.

L. 504-505: This last sentence does not seem to fit with the rest of the paragraph. I think that, with some rewriting, this could become clearer.

Technical comments

L. 48: 'Global climate models (GCMs) is . . .' -> are

L. 110-111: 'Only a few examples has . . .' -> have

L. 112-113: '. . . applying bias adjustment improve projections' -> improves

L. 142: the section marker should be corrected

L. 194: I can't find the source of this problem, should not be referenced with co-authors. The official webpage by Springer (https://link.springer.com/book/10.1007%2F978-1-4471-3675-0#about) only mentions one author (Stuart Coles) and there is no mention of other authors elsewhere in the book. So unless I'm missing something, I think the more correct reference is Coles (2001).

L. 232-243: 'Hosking and Wallis (1987) . . . warns . . . . Instead, he recommends . . .'. Shouldn't these sentences be plural, or are you referring to 'the paper' in these sentences instead of 'the authors'?

L. 254: 'Kallache et al. (2011) and Laflamme et al. (2016) applies' -> apply, as this verb is referring to multiple papers and authors.

L. 265: 'ths' -> 'the'

Figure 6 and Figure 8: Would it be possible to remove the underscores from the plot titles?

References

Cannon, A. J., Sobie, S. R., and Murdock, T. Q.: Bias correction of GCM precipitation by

quantile mapping: How well do methods preserve changes in quantiles and extremes?, Journal of Climate, 28, 6938–6959, https://doi.org/10.1175/JCLI-D-14-00754.1, 2015

Chen, J., Brissette, F. P., and Lucas-Picher, P.: Assessing the limits of bias-correcting climate model outputs for climate change impact studies, Journal of Geophysical Research: Atmospheres, 120, 1123–1136, https://doi.org/10.1002/2014JD022635, 2015

Hui, Y., Chen, J., Xu, C.-Y., Xiong, L., and Chen, H.: Bias nonstationarity of global climate model outputs: The role of internal climate variability and climate model sensitivity, International Journal of Climatology, 39, 2278–2294, https://doi.org/10.1002/joc.5950, 2019

Johnson, F. and Sharma, A.: Accounting for interannual variability: A comparison of options for water resources climate change impact assessments, Water Resources Research, 47, W04 508, https://doi.org/10.1029/2010WR009272, 2011

Kerkhoff, C., Künsch, H. R., and Schär, C.: Assessment of bias assumptions for climate models, Journal of Climate, 27, 6799–6818, https://doi.org/10.1175/JCLI-D-13-00716.1, 2014

Maraun, D. and Widmann, M.: Statistical Downscaling and Bias Correction for Climate Research, Cambridge University Press, https://doi.org/10.1017/9781107588783, 2018

Michelangeli, P.-A., Vrac, M., and Loukos, H.: Probabilistic downscaling approaches: Application to wind cumulative distribution functions, Geophysical Research Letters, 36, L11 708, https://doi.org/10.1029/2009GL038401, 2009

Van Schaeybroeck, B. and Vannitsem, S.: Assessment of calibration assumptions under strong climate changes, Geophysical Research Letters, 43, 1314–1322, https://doi.org/10.1002/2016GL067721, 2016

Velázquez, J. A., Troin, M., Caya, D., and Brissette, F.: Evaluating the time-invariance hypothesis of climate model bias correction: implications for hydrological impact studies, Journal of Hydrometeorology, 16, 2013–2026, https://doi.org/10.1175/JHM-D-14-

0159.1, 2015

Wang, Y., Sivandran, G., and Bielicki, J. M.: The stationarity of two statistical downscaling methods for precipitation under different choices of cross-validation periods, International Journal of Climatology, 38, e330–e348, https://doi.org/10.1002/joc.5375, 2018

---

## Short Comment (SC1) · 15 Sep 2020

**Timo Kelder**

t.kelder@lboro.ac.uk

Received and published: 15 September 2020

Comment on 'Identifying robust bias adjustment methods for extreme precipitation in a pseudo-reality setting' T. Kelder, R. L. Wilby, T. Marjoribanks, L. Slater

Torben Schmith and co-authors address a complex, but important topic. Climate model corrections typically assume stationary biases between simulated and observed extreme precipitation but, in practice, such biases may well be nonstationary (i.e. distributions may shift significantly in the future). Robust evaluation of bias correction methods is hampered by the inability to analyse future model biases, since there are obviously no observations of the future. To address this issue, the authors use model simulations as a pseudo-reality of the present and future climate to evaluate the robustness of various bias correction methods within these 'virtual' worlds.

The authors processed a large amount of data from the EURO-CORDEX ensemble and we commend them for this interesting research and their purposeful discussion of findings. The paper concludes by recommending a preferred bias correction method for climate projection. We offer a few suggestions and raise some issues for further elaboration by the authors.

1. Given that the analysis is based on an ensemble of climate model experiments, the logic should be explained for treating model-to-model biases in extreme precipitation as equivalent to model-to-observation biases. The paper acknowledges the limited ability of  $\sim$ 10km resolution model simulations at representing convective processes. Hence, more explanation is needed for an unfamiliar reader on why model experiments can be used to draw conclusions about the best bias correction methods on hourly timescales, if one cannot trust the model simulations to realistically represent convective processes.

2. Related to #1, a few cautionary remarks could be made about some of the GCMs used to drive the CORDEX experiments (see: Liepert and Lo, 2013). The realism of the downscaled extreme precipitation depends on the realism of the boundary forcing. Use of an 'ensemble of opportunity' is not unusual, but some studies narrow the choice of candidate models (and hence uncertainty) based on physical realism tests (e.g. McSweeney et al., 2015; Rowell, 2019).

3. In the inter-model cross-validation setup, every model/pseudo-reality combination is used. This setup can be useful for assessing relationships between present and future bias correction factors (e.g. Fig. 9), but does not mimic climate projections, where the ensemble mean, and range are typically used. In the present setup, a future projection is treated as a deterministic prediction, rather than a probabilistic projection. Perhaps use of the climate 'pseudo-observed' run might be favoured over future predictions simply because there is less variability in the present climate? How sensitive are the
results to taking the mean of all ensemble members minus the 'pseudo-reality' member (e.g. Fig. 3 in Räty et al. 2014)? This has the added benefit of involving much fewer permutations (and hence calculations).

4. The range of the projection matters. For example, Fig. 4 shows that there are future scenarios that exceed the present climate range. Hence, the worst-case 10-year precipitation event from the 'pseudo-obs' range would not include plausible future 10-year events. Therefore, more qualification is needed in the Abstract and Conclusions to guard against this possibility and the potentially misleading assertion that "the superior approach is to simply deduce future return levels from observations". Overall, the headline findings of the research could be presented in more nuanced ways, especially within the Abstract.

5. The Abstract and Introduction assert that "Severe precipitation events are usually projected using Regional Climate Model (RCM) scenario simulations." We gently remind the authors that statistical downscaling is also widely used for projecting severe precipitation events and suggest that more inclusive wording be used.

References Liepert, B.G. and F. Lo, 2013: CMIP5 update of 'Inter-model variability and biases of the global water cycle in CMIP3 coupled climate models'. Environ. Res. Lett., 8(2), p.029401, https://iopscience.iop.org/article/10.1088/1748-9326/8/2/029401/meta.

McSweeney, C.F., Jones, R.G., Lee, R.W. and Rowell, D.P., 2015: Selecting CMIP5 GCMs for downscaling over multiple regions. Clim. Dyn., 44(11), 3237-3260, https://doi.org/10.1007/s00382-014-2418-8.

Räty, O., J. Räisänen, and J. S. Ylhäisi, 2014: Evaluation of delta change and bias correction methods for future daily precipitation: intermodel cross-validation using EN-SEMBLES simulations. Clim. Dyn., 42, 2287–2303, https://doi.org/10.1007/s00382-014-2130-8.

Rowell, D.P., 2019: An observational constraint on CMIP5 projections of the East
African Long Rains and Southern Indian Ocean warming. Geophys. Res. Lett., 46(11), 6050-6058, https://doi.org/10.1029/2019GL082847.

---

## Referee Comment (RC3) · Anonymous Referee #3 · 18 Sep 2020

Overall comment

Overall, I recommend a better embedding of the manuscript in the current literature, both in introduction (e.g. much work has been done on comparing different bias correction methods, which could be included) and the section 5.1 could easily be expanded.

I also would like to see expansion on why different methods give different results. There seems to be no analysis or discussion of what features of different methods contribute to greater or lesser skill. In my view the manuscript would be improved if this were addressed.

Minor comments

105-106: It is true that future model performance cannot be tested directly. However,

split-sample testing is probably the best tool we have for this, particularly when a suspected climate change signal is present in recent historical data.

Figure 2,3: I find the colour scale used in these figure inappropriate. Yes, extreme precipitation events are projected to increase, but the scale make the increases look quite alarming. A percentage scale, and/or scale starting at zero would be more appropriate.

372-373, this sentence describing relative errors is a little unclear, I would suggest writing "Relative errors from the OBS method are in the range of 20%-40%" or similar.

395 and elsewhere: I'd use "percentiles" rather than "fractiles", e.g. 95th percentile rather than 0.95 fractile

The writing is generally of a high quality, but with a few corrections needed, such as:

48: "GCMs are"

182: "statistics are"

I recommend a thorough proofread to catch any other corrections

---

## Author Comment (AC2) · 14 Oct 2020

Author reply to:

Referee comment #2 on "Identifying robust bias adjustment methods for extreme precipitation in a pseudo-reality setting" by Torben Schmith et al.

We will start by thanking the referee for a fair and thorough review. We will comment (marked with »> . . . «<) on each review items below.

General comments

In their contribution, Schmith et al. (2020) discuss the robustness of different bias-adjusting methods for (sub)daily rainfall extremes. This yields interesting results and

strong links with the context of convection-permitting models and emergent constraints. Yet, there are some aspects about whom I'd like a deeper discussion.

»> we appreciate this positive overall judgement of our manuscript and are positive towards adding more discussion to it.«<

The first aspect is the practical use of this study. This is foremost linked with the choice of bias-adjusting methods. Although the use of return periods is perfectly justified from a hydrological point of view, I've seen few studies that actually use bias adjustment directly on the return periods. As such, I'd like to see a larger discussion on the choice of bias-adjusting methods.

»> Our aim has been to evaluate basic adjustment methods, used in hydrological applications. The simple climate factor approach has been applied in numerous hydrological applications, such as in (Sunyer et al. 2015; DeGaetano and Castellano 2017) and others. We also wanted to test quantile-mapping approaches, which in extreme value theory takes the form of a parametric transfer function. This we have applied in two flavours in the spirit of (Räty et al. 2014). Finally, we wanted to benchmark against gainst the 'canonical' benchmark methods, (observations and raw model output). We will discuss this in a revised manuscript.«<

Given a well-justified choice, I understand the use of these simple methods, yet I'd like to see more discussion on how this relates with more complicated, but related bias-adjustment methods, such as e.g. CDF-t (Michelangeli et al., 2009), standard QM, QDM (Cannon et al., 2015), : : : Would it be possible to discuss possible consequences for the use of these methods for the adjustment of subdaily precipitation extremes? This could fit in the second paragraph of Section 5.1, which seems rather limited and abrupt at this point.

»> We will add some discussion of the more elaborate quantile mapping methods. Also discuss expected consequences for the adjustment, if possible. We will, however, emphasize that these methods build on alternative, but not necessarily more correct,

assumptions. It would be interesting to test these methods in our framework, but we reserve this to future publications. In this connection, not that our investigation do not generally find that the more advanced methods (quantile mapping) outperform the simpler climate factor approach.«<

A last point related to the practical use is that I missed a more thorough explanation of why the observations perform well, why this version of quantile mapping performs poorly. Although this is discussedcslightly in Section 4.3, I wonder if more details or, if possible, practical guidelines could be given in the discussion.

»>A thorough reveal of causes for some models performing well would require quite some extra analysis which cannot be accommodated within this manuscript. We may speculate that the cause of observations performing so well as projection is related to the poor signal-to-noise ratio, as seen in Fig. 4. The relatively poor performance of the quantile-matching methods could be caused by the many extreme value distributions to be estimated, each of which are very uncertain.«<

A second aspect is that some concepts in the Introduction seem to be accepted as-is, whereas they could deserve a deeper discussion. A first example of this is the discussion of stationarity in the introduction. The references are limited in time, whereas more recent papers expanded this subject, such as Kerkhoff et al. (2014) and Van Schaeybroeck and Vannitsem (2016) on the type of bias relationship and Chen et al. (2015), Velázquez et al. (2015), Wang et al. (2018) and Hui et al. (2019), who discussed the uncertainty introduced by bias nonstationarity. As the stationarity of the bias is an important part of the discussion, I think the paper could benefit from these perspectives.

»> We were not aware of the above references, forwarding interesting aspect of stationarity. It is relevant to discuss this in the introduction, and we will do so in a modified manuscript.«<

A second, smaller example is the use of a delta change based method. While the
method isn't completely discredited, there has been some discussion whether it's use for climate change is not too dependent on the assumption that the temporal structure of the time series will not change from present to future (e.g. Johnson and Sharma (2011), Kerkhoff et al. (2014)). It would thus be interesting to read a deeper discussion on the limitations of the methods

»>We are aware of the assumption about unchanged temporal structure of time series in the delta change approach. Even if so, delta change methods were included in the studies (Räty et al. 2014; Räisänen and Räty 2013), and therefore we choose to include such methods as well. «<

Specific comments

L. 37: 'quantile-mapping' is used here, whereas in the remainder of the abstract (and the paper) 'quantile-matching' is used. I'd suggest to edit this for coherence, but to also use 'quantile mapping' throughout the paper, as it has been the most used term for this type of bias adjustment during the last few years.

»> Certainly, the nomenclature should be consistent throughout. We well take care of that, and follow your advice, adhering to the term 'quantile mapping' «<

L. 75-82: this paragraph is very scarce on references. Although some of the necessary references are given in the discussion, I think it would be good to also have the reference to the papers about CPMs in this paragraph.

»> Ok, we will do so.«<

L. 84-91: The terminology in this paragraph could be reconsidered. Although it is debatable whether or not to consider delta change as a bias adjustment approach (the latest textbook, Maraun and Widmann (2018), is on the edge), it feels very strange to read 'bias correction' as a subset of 'bias adjustment' approaches. The use of 'bias adjustment' as a replacement of 'bias correction' has been rising during the last few years, as it is clearer that the methods are statistical and cannot correct all climate

model biases. Thus, I would withhold from the use of 'bias correction'. Better terminology seems MOS, with delta change and bias adjustment as possible subcategories, or bias adjustment with delta change and bias adjustment s.s., although the exact choice is personal.

»> Yes, exactly this has also been on our minds! We think that your suggestion of using the generic term 'MOS' would be a choice to apply.«<

L. 253- 286: Although the method described here is indeed based on the same principles as XCDF-t as used by Kallache et al. (2011) and Laflamme et al. (2016), it's not entirely clear how the new method is created by adapting the former. I think the link between both methods should be more detailed, so users can retrace it more easily and infer the strengths and limitations. Especially as it is specifically mentioned that the method 'will be adapted to our needs below', the adaptation seems rather limited.

»> We will make the connection with XCDF-t more clear in the revised manuscript.

L. 448-453: the explanation of the use of the index by Maurer et al. (2013) should be expanded. Firstly, it's unclear to me where the terminology 'measure of relative spread' is derived from, as it is not named as such in the original paper. Secondly, the interpretation of the R-values is not discussed, although this is quite important: values < 1 indicate that the difference in biases is smaller than the mean bias of both periods, whereas values >1 indicate that the difference in biases is larger, which could have a potentially large impact. As both values are quite far < 1, the bias seems quite stationary, but in your discussion you state that the 24h duration is 'less stationary'. Without giving this numerical explanation, this statement is hard to interpret correctly.

»> We will expand the explanation of R, and its interpretation, as suggested. Certainly, both R-values are below 1. However, R=0 is a sign of a stationary bias factor and this is the basis of our interpretation«<

L. 504-505: This last sentence does not seem to fit with the rest of the paragraph. I

think that, with some rewriting, this could become clearer.

»>We will rephrase this sentence.«<

Technical comments

»> we will adhere to the technical comments given below«<

L. 48: 'Global climate models (GCMs) is : : :' -> are

L. 110-111: 'Only a few examples has : : :' -> have

L. 112-113: ': : : applying bias adjustment improve projections' -> improves

L. 142: the section marker should be corrected

L. 194: I can't find the source of this problem, should not be referenced with co-authors. The official webpage by Springer (https://link.springer.com/book/10.1007%2F978-1-4471-3675-0#about) only mentions one author (Stuart Coles) and there is no mention of other authors elsewhere in the book. So unless I'm missing something, I think the more correct reference is Coles (2001).

L. 232-243: 'Hosking and Wallis (1987) : : : warns : : : . Instead, he recommends : : :'. Shouldn't these sentences be plural, or are you referring to 'the paper' in these sentences instead of 'the authors'?

L. 254: 'Kallache et al. (2011) and Laflamme et al. (2016) applies' -> apply, as this verb is referring to multiple papers and authors.

L. 265: 'ths' -> 'the'

Figure 6 and Figure 8: Would it be possible to remove the underscores from the plot titles?

References

Cannon, A. J., Sobie, S. R., and Murdock, T. Q.: Bias correction of GCM precipitation by

quantile mapping: How well do methods preserve changes in quantiles and extremes?, Journal of Climate, 28, 6938–6959, https://doi.org/10.1175/JCLI-D-14-00754.1, 2015

Chen, J., Brissette, F. P., and Lucas-Picher, P.: Assessing the limits of bias-correcting climate model outputs for climate change impact studies, Journal of Geophysical Research: Atmospheres, 120, 1123–1136, https://doi.org/10.1002/2014JD022635, 2015

Hui, Y., Chen, J., Xu, C.-Y., Xiong, L., and Chen, H.: Bias nonstationarity of global climate model outputs: The role of internal climate variability and climate model sensitivity, International Journal of Climatology, 39, 2278–2294, https://doi.org/10.1002/joc.5950, 2019

Johnson, F. and Sharma, A.: Accounting for interannual variability: A comparison of options for water resources climate change impact assessments, Water Resources Research, 47, W04 508, https://doi.org/10.1029/2010WR009272, 2011

Kerkhoff, C., Künsch, H. R., and Schär, C.: Assessment of bias assumptions for climate models, Journal of Climate, 27, 6799–6818, https://doi.org/10.1175/JCLI-D-13-00716.1, 2014

Maraun, D. and Widmann, M.: Statistical Downscaling and Bias Correction for Climate Research, Cambridge University Press, https://doi.org/10.1017/9781107588783, 2018

Michelangeli, P.-A., Vrac, M., and Loukos, H.: Probabilistic downscaling approaches: Application to wind cumulative distribution functions, Geophysical Research Letters, 36, L11 708, https://doi.org/10.1029/2009GL038401, 2009

Van Schaeybroeck, B. and Vannitsem, S.: Assessment of calibration assumptions under strong climate changes, Geophysical Research Letters, 43, 1314–1322, https://doi.org/10.1002/2016GL067721, 2016

Velázquez, J. A., Troin, M., Caya, D., and Brissette, F.: Evaluating the time-invariance hypothesis of climate model bias correction: implications for hydrological impact studies, Journal of Hydrometeorology, 16, 2013–2026, https://doi.org/10.1175/JHM-D-14-

C50159.1, 2015

Wang, Y., Sivandran, G., and Bielicki, J. M.: The stationarity of two statistical down-scaling methods for precipitation under different choices of cross-validation periods, International Journal of Climatology, 38, e330–e348, https://doi.org/10.1002/joc.5375, 2018

Our added references:

DeGaetano, A. T., and C. M. Castellano, 2017: Future projections of extreme precipitation intensity-duration-frequency curves for climate adaptation planning in New York State. Clim. Serv., 5, 23–35, https://doi.org/10.1016/j.cliser.2017.03.003.

Räisänen, J., and O. Räty, 2013: Projections of daily mean temperature variability in the future: cross-validation tests with ENSEMBLES regional climate simulations. Clim. Dyn., 41, 1553–1568, https://doi.org/10.1007/s00382-012-1515-9.

Räty, O., J. Räisänen, and J. S. Ylhäisi, 2014: Evaluation of delta change and bias correction methods for future daily precipitation: intermodel cross-validation using EN-SEMBLES simulations. Clim. Dyn., 42, 2287–2303, https://doi.org/10.1007/s00382-014-2130-8.

Sunyer, M. A., I. B. Gregersen, D. Rosbjerg, H. Madsen, J. Luchner, and K. Arnbjerg-Nielsen, 2015: Comparison of different statistical downscaling methods to estimate changes in hourly extreme precipitation using RCM projections from ENSEMBLES. Int. J. Climatol., 35, 2528–2539, https://doi.org/10.1002/joc.4138.

---

## Author Comment (AC3) · 14 Oct 2020

Author reply to:

Short comment #1 on "Identifying robust bias adjustment methods for extreme precipitation in a pseudo-reality setting" by Torben Schmith et al.

We will start by thanking our collegues in science for this short comment. We will comment (marked with »> . . . «<) on each of their items below.

Comment on 'Identifying robust bias adjustment methods for extreme precipitation in a pseudo-reality setting' T. Kelder, R. L. Wilby, T. Marjoribanks, L. Slater

Torben Schmith and co-authors address a complex, but important topic. Climate model

corrections typically assume stationary biases between simulated and observed extreme precipitation but, in practice, such biases may well be nonstationary (i.e. distributions may shift significantly in the future). Robust evaluation of bias correction methods is hampered by the inability to analyse future model biases, since there are obviously no observations of the future. To address this issue, the authors use model simulations as a pseudo-reality of the present and future climate to evaluate the robustness of various bias correction methods within these 'virtual' worlds. The authors processed a large amount of data from the EURO-CORDEX ensemble and we commend them for this interesting research and their purposeful discussion of findings. The paper concludes by recommending a preferred bias correction method for climate projection. We offer a few suggestions and raise some issues for further elaboration by the authors.

1. Given that the analysis is based on an ensemble of climate model experiments, the logic should be explained for treating model-to-model biases in extreme precipitation as equivalent to model-to-observation biases. The paper acknowledges the limited ability of _10km resolution model simulations at representing convective processes. Hence, more explanation is needed for an unfamiliar reader on why model experiments can be used to draw conclusions about the best bias correction methods on hourly timescales, if one cannot trust the model simulations to realistically represent convective processes.

»>Our approach of treating model-to-model biases as equivalent to model-to-observations is named 'indistinguishable interpretation', as opposed to the 'truth plus error interpretation' (Sanderson and Knutti 2012). Our motivation of adopting the former is indirect. For variables, such as the large-scale surface temperature, which are used as measure in the tuning process of models, the truth plus error is the canonical choice. Precipitation is not directly linked to the tuning process and has smaller scale, and as such the indistinguishable interpretation can be argued for (Christiansen 2020).

Acknowledging that the models represent convection imperfect, we are actually better

off evaluating the bias correction methods between models than between model and observation. We are here addressing the statistical nature of the corrections, not the physical processes which bias correction methods are not suitable for anyway. We do not promote, naively applying these methods to hourly data from these models. However, the presented methods can in the future be applied to convection permitting model simulations that better represent the convective process, and results from our current manuscript would apply equally to that case. We do not presently have a large ensemble available of such models to perform our study on. We will explain this view in a revised manuscript.«<

2. Related to #1, a few cautionary remarks could be made about some of the GCMs used to drive the CORDEX experiments (see: Liepert and Lo, 2013). The realism of the downscaled extreme precipitation depends on the realism of the boundary forcing. Use of an 'ensemble of opportunity' is not unusual, but some studies narrow the choice of candidate models (and hence uncertainty) based on physical realism tests (e.g. McSweeney et al., 2015; Rowell, 2019).

»>We only partly agree with this. The large-scale atmospheric state is certainly determined by the boundary forcing; though, the RCM is able to modulate it. Distribution of precipitation intensities are to a large extent determined by the RCM (see e.g. (Christensen and Kjellström 2020)). This is particularly true for the high-extreme end of the spectrum.

We are aware of the use of selection procedures put forward in the cited papers. There is, however, no simple quality index that can be generally applied. Any discrimination of GCMs depends depend on area, season, and the meteorological field and property being investigated (Gleckler et al. 2008); e.g. their Fig. 9). Furthermore, these tests and selection procedures are based on subjective criteria and come with major caveats that impact the uncertainty range largely (Madsen et al. 2017). We therefore choose, in accordance with most other similar studies, to use 'ensemble of opportunity' for the present study.«<

3. In the inter-model cross-validation setup, every model/pseudo-reality combination is used. This setup can be useful for assessing relationships between present and future bias correction factors (e.g. Fig. 9), but does not mimic climate projections, where the ensemble mean, and range are typically used. In the present setup, a future projection is treated as a deterministic prediction, rather than a probabilistic projection. Perhaps use of the climate 'pseudo-observed' run might be favoured over future predictions simply because there is less variability in the present climate? How sensitive are the results to taking the mean of all ensemble members minus the 'pseudo-reality' member (e.g. Fig. 3 in Räty et al. 2014)? This has the added benefit of involving much fewer permutations (and hence calculations).

»>Actually a good idea, which we have implemented in our analysis suite (see attached plots). We think we can find the space for two extra plots and some associated text in the revised version. «<

4. The range of the projection matters. For example, Fig. 4 shows that there are future scenarios that exceed the present climate range. Hence, the worst-case 10-year precipitation event from the 'pseudo-obs' range would not include plausible future 10-year events. Therefore, more qualification is needed in the Abstract and Conclusions to guard against this possibility and the potentially misleading assertion that "the superior approach is to simply deduce future return levels from observations". Overall, the headline findings of the research could be presented in more nuanced ways, especially within the Abstract.

»>We are afraid that we do not understand the central statement of this point ("Hence, the worst-case . . .). Therefore, we are not able to comment on it.«<

5. The Abstract and Introduction assert that "Severe precipitation events are usually projected using Regional Climate Model (RCM) scenario simulations." We gently remind the authors that statistical downscaling is also widely used for projecting severe precipitation events and suggest that more inclusive wording be used.

»>We agree that this suggestion is appropriate. It could easily be met by referring to work by Wilby, Mauraun, and others.«<

References

Liepert, B.G. and F. Lo, 2013: CMIP5 update of 'Inter-model variability and biases of the global water cycle in CMIP3 coupled climate models'. Environ. Res. Lett., 8(2), p.029401, https://iopscience.iop.org/article/10.1088/1748-9326/8/2/029401/meta.

McSweeney, C.F., Jones, R.G., Lee, R.W. and Rowell, D.P., 2015: Selecting CMIP5 GCMs for downscaling over multiple regions. Clim. Dyn., 44(11), 3237-3260, https://doi.org/10.1007/s00382-014-2418-8.

Räty, O., J. Räisänen, and J. S. Ylhäisi, 2014: Evaluation of delta change and bias correction methods for future daily precipitation: intermodel cross-validation using EN-SEMBLES simulations. Clim. Dyn., 42, 2287–2303, https://doi.org/10.1007/s00382-014-2130-8.

Rowell, D.P., 2019: An observational constraint on CMIP5 projections of the East African Long Rains and Southern Indian Ocean warming. Geophys. Res. Lett., 46(11), 6050-6058, https://doi.org/10.1029/2019GL082847.

»>Our added references:

Christensen, O. B., and E. Kjellström, 2020: Partitioning uncertainty components of mean climate and climate change in a large ensemble of European regional climate model projections. Clim. Dyn., 54, 4293–4308, https://doi.org/10.1007/s00382-020-05229-y.

Christiansen, B., 2020: Understanding the Distribution of Multimodel Ensembles. J. Clim., 33, 9447–9465, https://doi.org/10.1175/JCLI-D-20-0186.1.

Gleckler, P. J., K. E. Taylor, and C. Doutriaux, 2008: Performance metrics for climate models. J. Geophys. Res., 113, D06104, https://doi.org/10.1029/2007JD008972.

Madsen, M. S., P. L. Langen, F. Boberg, and J. H. Christensen, 2017: Inflated Uncertainty in Multimodel‐Based Regional Climate Projections. Geophys. Res. Lett., 44, 11,606-11,613, https://doi.org/10.1002/2017GL075627.

Sanderson, B. M., and R. Knutti, 2012: On the interpretation of constrained climate model ensembles. Geophys. Res. Lett., 39, n/a-n/a, https://doi.org/10.1029/2012GL052665. «<
[Figure]

[Figure]

Relative error (ensemble mean), Duration: 1 h

Fig. 1.

[Figure]

Fig. 2.

---

## Author Comment (AC4) · 14 Oct 2020

Referee comment #3 on "Identifying robust bias adjustment methods for extreme precipitation in a pseudo-reality setting" by Torben Schmith et al.

We will start by thanking the referee for a fair review. We will comment (marked with »> . . . «<) on each review items below.

Overall comment

Overall, I recommend a better embedding of the manuscript in the current literature, both in introduction (e.g. much work has been done on comparing different bias correction methods, which could be included) and the section 5.1 could easily be expanded. I also would like to see expansion on why different methods give different results. There

seems to be no analysis or discussion of what features of different methods contribute to greater or lesser skill. In my view the manuscript would be improved if this were addressed.

»>We will meet the advice of a more thourough embedding in the relevant. This will be followed by adhering to the y in particular referee #2. To disentangle why different methods give different results requires more analysis requires extensive analysis and has to be left to future work. We have given an appetizer of this kind of work in section 4.3.«<

Minor comments

105-106: It is true that future model performance cannot be tested directly. However, split-sample testing is probably the best tool we have for this, particularly when a suspected climate change signal is present in recent historical data.

»>as we see it, split-sample testing is an alternative to our approach. We will be happy to receive any arguments, why it should be the best tool.«<

»>We will incorporate all followings remarks about figure layout and wording into a revised manuscript:«<

Figure 2,3: I find the colour scale used in these figure inappropriate. Yes, extreme precipitation events are projected to increase, but the scale make the increases look quite alarming. A percentage scale, and/or scale starting at zero would be more appropriate.

372-373, this sentence describing relative errors is a little unclear, I would suggest writing "Relative errors from the OBS method are in the range of 20%-40%" or similar.

395 and elsewhere: I'd use "percentiles" rather than "fractiles", e.g. 95th percentile rather than 0.95 fractile

The writing is generally of a high quality, but with a few corrections needed, such as: 48: "GCMs are" 182: "statistics are" I recommend a thorough proofread to catch any

other corrections

---

## Author Response (AR1)

Dear editor,

We appreciate very much the comment from the referees and from our collegues posting a SC. These have been extremely useful for improving the manuscript. Below, you find all comments comments with our responses in *italic*. Section numbers and line numbers refer to the marked-up version of the revised manuscript.

**Anonymous Referee #1:**

This is an interesting contribution involving a lot of work. I have a few general issues that the authors should address in their revisions, followed by some specific comments. Firstly - there needs to be a better discussion about the possible problems in using the pseudo-reality setting for assessment of precipitation extremes. Most models have a tendency to increase the probability of occurrence of rainfall, thereby increasing the size of the sample that could potentially constitute extremes. The authors have avoided this issue to some extent by performing a pseudo-reality assessment. I believe some discussion should be included as this could create difficulties in taking the findings from here to real applications.

> *We have added/modified the intro (lines 172-180) about different validation approaches and their pros and con's. We recognize that models do have a tendency to increased probability of rainfall. As for the last part of the comment, we determine our POT threshold by having three events/year instead of having a fixed threshold. Therefore, we always have the same pool of extremes, regardless of model and present-day/end-21$^{st}$-century.*

Secondly, the paper is coming across as a bit of a report (and I sympathise with the authors as they do have a lot of information to present). Perhaps a more creative discussion for differences in mountaineous areas versus not, coastal areas versus not, and daily durations versus hourly would be useful. I note the spatial resolution is 11km. Daily extremes should be simulated better at this resolution.

> *Thanks for the advice. We have worked through the text and realize that maybe you think of Section 4.1. Therefore, we have extended the description of Figures 2 and 3. Furthermore, these figures have been modified, caused by a suggestion from another referee.*

Also, no mention is made of the causative GCMs that are interpolated using the RCMs. There are different extent of biases in these. Some discussion should be included on this as well.

> *We have introduced some text on this in section 2, mentioning good performance of GCMs and the argument for using 'ensemble of opportunity' in favour of selection procedures.*

Thirdly, the authors have missed with publications on this topic by Jingwan Li. Relevant papers are: Li, J., et al. (2017). "A comparison of methods for estimating climate change impact on design rainfall using a high-resolution RCM." Journal of Hydrology 547: 413-427. Li, J., et al. (2017). "A comparison of methods to estimate future subdaily design rainfall." Advances in Water Resources 110: 215-227. Li, J., et al. (2018). "Addressing the mischaracterization of extreme rainfall in regional climate model simulations – A synoptic pattern based bias correction approach." Journal of Hydrology 556: 901-912. Li, J., et al. (2018). "Can Regional Climate Modeling Capture the Observed Changes in Spatial Organization of Extreme Storms at Higher Temperatures?" Geophysical Research Letters 45(9): 4475-4484.

I am a co-author on these papers hence have a conflict here. But I think these are very relevant to what the authors are attempting to do here, as she used an even finer resolution RCM with a high density of observed gauges at the same time resolution (hourly). The bias correction approach she adopted acknowledged the bias in simulating convection within the RCMs as well as the quantile bias convective and non-convective rainfall were exhibiting.

> *We were not aware of these papers. We are now referring to the two papers "A comparison …" in the introduction (line 158). Our manuscript evaluates basic adjustment methods only. We know that there is a myriad of special-designed adjust methods, including the one described in the paper "Addressing the mischaracterization … ". We have added a section (5.5, lines 688-714) discussing which methods were/were not included in our study. The paper "Can Regional Climate Modeling Capture …" about the spatial extent of extreme precipitation events is in our opinion not within the scope of our manuscript.*

Now to the specific comments:

line 142 - missing section marker

> *Thanks, has been fixed.*

line 225 - there is another way to create the partial series sample. It is to acknowledge that there may be a bias in the proportion of events that are say convective. If this proportion is biased, one is forming a biased sample effectively by selecting the series the way adopted here. This issue is the focus of Li, J., et al. (2018). "Addressing the mischaracterization of extreme rainfall in regional climate model simulations – A synoptic pattern based bias correction approach." Journal of Hydrology 556: 901-912.

> *In the manuscript we evaluate the basic methods (see line 690). The work described in the suggested paper is not within our scope (see also above).*

line497 - If the proportion of convective extreme events increases in the future (as it is expected to) then ignoring any bias in the representation of convection as discussed above, will create a non-stationary bias. This can be addressed though using the above mentioned approach.

*The aim of our work is to evaluate the simple bias adjustment methods for extremes, as also explained above. More sophisticated methods are not included in this study, but the suggested paper can go into the discussion on future work.*

**Referee #2:**

General comments

In their contribution, Schmith et al. (2020) discuss the robustness of different bias-adjusting methods for (sub)daily rainfall extremes. This yields interesting results and strong links with the context of convection-permitting models and emergent constraints. Yet, there are some aspects about whom I'd like a deeper discussion.

*We appreciate this positive overall judgement of our manuscript and are positive towards adding more discussion to it.*

The first aspect is the practical use of this study. This is foremost linked with the choice of bias-adjusting methods. Although the use of return periods is perfectly justified from a hydrological point of view, I've seen few studies that actually use bias adjustment directly on the return periods. As such, I'd like to see a larger discussion on the choice of bias-adjusting methods.

*Our aim has been to evaluate basic adjustment methods. We have added a new subsection in the discussion (lines 588-714) summarizing the more elaborate quantile mapping methods.*

Given a well-justified choice, I understand the use of these simple methods, yet I'd like to see more discussion on how this relates with more complicated, but related bias-adjustment methods, such as e.g. CDF-t (Michelangeli et al., 2009), standard QM, QDM (Cannon et al., 2015), : : : Would it be possible to discuss possible consequences for the use of these methods for the adjustment of subdaily precipitation extremes? This could fit in the second paragraph of Section 5.1, which seems rather limited and abrupt at this point.

*In a new sub-section (lines 588-714) we discuss the use of more elaborate methods. We emphasize that these methods build on alternative, but not necessarily more correct, assumptions. It would be interesting to test these methods in our framework, but we reserve this to future publications. We also note that our investigation do not generally find that the more elaborate methods (quantile mapping) outperform the simpler climate factor approach.*

A last point related to the practical use is that I missed a more thorough explanation of why the observations perform well, why this version of quantile mapping performs poorly. Although this is discussed slightly in Section 4.3, I wonder if more details or, if possible, practical guidelines could be given in the discussion.

*A thorough reveal of causes for some models performing well would require quite some extra analysis which cannot be accommodated within this manuscript. We may speculate that the cause*

*of observations performing so well as projection is related to the poor signal-to-noise ratio, as seen in Fig. 4. The relatively poor performance of the quantile-matching methods could be caused by the many extreme value distributions to be estimated, each of which are very uncertain. We have added a block of text on this in the Conclusions section.*

A second aspect is that some concepts in the Introduction seem to be accepted as-is, whereas they could deserve a deeper discussion. A first example of this is the discussion of stationarity in the introduction. The references are limited in time, whereas more recent papers expanded this subject, such as Kerkhoff et al. (2014) and Van Schaeybroeck and Vannitsem (2016) on the type of bias relationship and Chen et al. (2015), Velázquez et al. (2015), Wang et al. (2018) and Hui et al. (2019), who discussed the uncertainty introduced by bias nonstationarity. As the stationarity of the bias is an important part of the discussion, I think the paper could benefit from these perspectives.

*In the original submitted manuscript, stationarity was mentioned and briefly discussed in the introduction. We have written a new discussion and updated the references (lines 136-147).*

A second, smaller example is the use of a delta change based method. While the method isn't completely discredited, there has been some discussion whether it's use for climate change is not too dependent on the assumption that the temporal structure of the time series will not change from present to future (e.g. Johnson and Sharma (2011), Kerkhoff et al. (2014)). It would thus be interesting to read a deeper discussion on the limitations of the methods

*We are aware of the assumption about unchanged temporal structure of time series in the delta change approach, though this is only 100% true in the simplest version of a shift of the mean, in the quantile mapping version temporal structure may change. Furthermore, our MOS of extreme levels do not yield any time series as output. Therefore, we think that a discussion as suggested is not relevant for our manuscript.*

Specific comments

L. 37: 'quantile-mapping' is used here, whereas in the remainder of the abstract (and the paper) 'quantile-matching' is used. I'd suggest to edit this for coherence, but to also use 'quantile mapping' throughout the paper, as it has been the most used term for this type of bias adjustment during the last few years.

*Certainly, the nomenclature should be consistent throughout. We have followed your advice and replaced 'quantile matching' to 'quantile mapping' throughout.*

L. 75-82: this paragraph is very scarce on references. Although some of the necessary references are given in the discussion, I think it would be good to also have the reference to the papers about CPMs in this paragraph.

*Ok, we have introduced the appropriate references*

L. 84-91: The terminology in this paragraph could be reconsidered. Although it is debatable whether or not to consider delta change as a bias adjustment approach (the latest textbook, Maraun and Widmann (2018), is on the edge), it feels very strange to read 'bias correction' as a subset of 'bias adjustment' approaches. The use of 'bias adjustment' as a replacement of 'bias correction' has been rising during the last few years, as it is clearer that the methods are statistical and cannot correct all climate model biases. Thus, I would withhold from the use of 'bias correction'. Better terminology seems MOS, with delta change and bias adjustment as possible subcategories, or bias adjustment with delta change and bias adjustment s.s., although the exact choice is personal.

> *It is indeed difficult to find a coherent terminology - with Maraun&Widman, there is a 'Babylonian confusion'. We have decided to use the generic term 'adjustment' (sometimes bias adjustment' to prevent confusion) with sub-categories 'bias correction' and 'delta change' throughout the revised manuscript. In the main headline, though, we keep 'bias correction' as the generic term for better readability.*

L. 253- 286: Although the method described here is indeed based on the same principles as XCDF-t as used by Kallache et al. (2011) and Laflamme et al. (2016), it's not entirely clear how the new method is created by adapting the former. I think the link between both methods should be more detailed, so users can retrace it more easily and infer the strengths and limitations. Especially as it is specifically mentioned that the method 'will be adapted to our needs below', the adaptation seems rather limited.

> *Our method was originally inspired by XCDF-t, but we make the more direct approach and define transformations, which are the used to correct the return levels. To avoid any confusion, we have chosen to remove the first lines of section 3.3.2*

L. 448-453: the explanation of the use of the index by Maurer et al. (2013) should be expanded. Firstly, it's unclear to me where the terminology 'measure of relative spread' is derived from, as it is not named as such in the original paper. Secondly, the interpretation of the R-values is not discussed, although this is quite important: values < 1 indicate that the difference in biases is smaller than the mean bias of both periods, whereas values >1 indicate that the difference in biases is larger, which could have a potentially large impact. As both values are quite far < 1, the bias seems quite stationary, but in your discussion you state that the 24h duration is 'less stationary'. Without giving this numerical explanation, this statement is hard to interpret correctly.

> *We have expanded the explanation of R, and its interpretation, as suggested. Certainly, both R-values are below 1. However, it is the limit of R=0 which is a sign of a stationary bias factor and this is the basis of our interpretation and discussion.*

L. 504-505: This last sentence does not seem to fit with the rest of the paragraph. I think that, with some rewriting, this could become clearer.

> *This reference doesn't really belong here, so we have deleted this sentence.*

Technical comments

*we will adhere to the technical comments given below*

L. 48: 'Global climate models (GCMs) is : : :' -> are      *done*

L. 110-111: 'Only a few examples has : : :' -> have      *done*

L. 112-113: ': : : applying bias adjustment improve projections' -> improves   *done*

L. 142: the section marker should be corrected      *ok*

L. 194: I can't find the source of this problem, should not be referenced with co-authors. The official webpage by Springer (https://link.springer.com/book/10.1007%2F978-1-4471-3675-0#about) only mentions one author (Stuart Coles) and there is no mention of other authors elsewhere in the book. So unless I'm missing something, I think the more correct reference is Coles (2001). *Yes, correct, has been changed.*

L. 232-243: 'Hosking and Wallis (1987) : : : warns : : : . Instead, he recommends : : :'. Shouldn't these sentences be plural, or are you referring to 'the paper' in these sentences instead of 'the authors'? *Probably one should refer to the authors, we have corrected*

L. 254: 'Kallache et al. (2011) and Laflamme et al. (2016) applies' -> apply, as this verb is referring to multiple papers and authors. *done*

L. 265: 'ths' -> 'the' *done*

Figure 6 and Figure 8: Would it be possible to remove the underscores from the plot titles? *Done*

**Referee #3**

Overall comment

Overall, I recommend a better embedding of the manuscript in the current literature, both in introduction (e.g. much work has been done on comparing different bias correction methods, which could be included) and the section 5.1 could easily be expanded. I also would like to see expansion on why different methods give different results. There seems to be no analysis or discussion of what features of different methods contribute to greater or lesser skill. In my view the manuscript would be improved if this were addressed.

> *We will meet this advice of a more thourough embedding in the relevant. This will be followed by adhering to suggestions given by in particular referee #2. To disentangle why different methods give different results requires more analysis requires extensive analysis and has to be left to future work. We have given an appetizer of this kind of work in section 4.3.*

Minor comments

105-106: It is true that future model performance cannot be tested directly. However, split-sample testing is probably the best tool we have for this, particularly when a suspected climate change signal is present in recent historical data.

> *as we see it, split-sample testing is an alternative to our approach; not neccesarily the best one. We have included a paragraph in the introduction discussing different validation approaches and their pros and con's (in lines 172-180), in accordance with suggestion from referee #1.*

Figure 2,3: I find the colour scale used in these figure inappropriate. Yes, extreme precipitation events are projected to increase, but the scale make the increases look quite alarming. A percentage scale, and/or scale starting at zero would be more appropriate.

*We have reacted to this piece of advice by showing instead maps of present-day and maps of the relative change*

372-373, this sentence describing relative errors is a little unclear, I would suggest writing "Relative errors from the OBS method are in the range of 20%-40%" or similar.

*Done*

and elsewhere: I'd use "percentiles" rather than "fractiles", e.g. 95th percentile rather than 0.95 fractile

*We agree that percentile is more widely used according to Google; therefore we have followed this advice. We have also changed all relative measures to percent throughout.*

The writing is generally of a high quality, but with a few corrections needed, such as:
48: "GCMs are"  *yes, thank you*
182: "statistics are"  *yes, thank you*
I recommend a thorough proofread to catch any other corrections

**Short comment #1**

Comment on 'Identifying robust bias adjustment methods for extreme precipitation in a pseudo-reality setting' T. Kelder, R. L. Wilby, T. Marjoribanks, L. Slater

Torben Schmith and co-authors address a complex, but important topic. Climate model corrections typically assume stationary biases between simulated and observed extreme precipitation but, in practice, such biases may well be nonstationary (i.e. distributions may shift significantly in the future). Robust evaluation of bias correction methods is hampered by the inability to analyse future model biases, since there are obviously no observations of the future. To address this issue, the authors use model simulations as a pseudo-reality of the present and future climate to evaluate the robustness of various bias correction methods within these 'virtual' worlds.
The authors processed a large amount of data from the EURO-CORDEX ensemble and we commend them for this interesting research and their purposeful discussion of findings. The paper concludes by recommending a preferred bias correction method for climate projection. We offer a few suggestions and raise some issues for further elaboration by the authors.

1. Given that the analysis is based on an ensemble of climate model experiments, the logic should be explained for treating model-to-model biases in extreme precipitation as equivalent to model-to-observation biases. The paper acknowledges the limited ability of _10km resolution model simulations at representing convective processes. Hence, more explanation is needed for an unfamiliar reader on why model experiments can be used to draw conclusions about the best bias correction methods on hourly timescales, if one cannot trust the model simulations to realistically represent convective processes.

*Acknowledging that the models represent convection imperfect, we are actually better off evaluating the bias correction methods between models than between model and observation. We are here addressing the statistical nature of the corrections, not the physical processes which bias correction methods are not suitable for anyway. We do not promote, naively applying these methods to hourly data from these models. However, the presented methods can in the future be applied to convection permitting model simulations that better represent the convective process, and results from our current manuscript would apply equally to that case. We have added a sentence about this in lines 636-640 of the revised manuscript.*

2. Related to #1, a few cautionary remarks could be made about some of the GCMs used to drive the CORDEX experiments (see: Liepert and Lo, 2013). The realism of the downscaled extreme precipitation depends on the realism of the boundary forcing. Use of an 'ensemble of opportunity' is not unusual, but some studies narrow the choice of candidate models (and hence uncertainty) based on physical realism tests (e.g. McSweeney et al., 2015; Rowell, 2019).

*We only partly agree with this. The large-scale atmospheric state is certainly determined by the boundary forcing; though, the RCM is able to modulate it. Distribution of precipitation intensities are to a large extent determined by the RCM (see e.g. (Christensen and Kjellström 2020)). This is particularly true for the high-extreme end of the spectrum.*

*We are aware of the use of selection procedures put forward in the cited papers. There is, however, no simple quality index that can be generally applied. Any discrimination of GCMs depends depend on area, season, and the meteorological field and property being investigated (Gleckler et al. 2008; e.g. their Fig. 9). Furthermore, these tests and selection procedures are based on subjective criteria and come with major caveats that impact the uncertainty range largely (Madsen et al. 2017). We therefore choose, in accordance with most other similar studies, to use 'ensemble of opportunity' for the present study. We now discuss that in lines 235-243.*

3. In the inter-model cross-validation setup, every model/pseudo-reality combination is used. This setup can be useful for assessing relationships between present and future bias correction factors (e.g. Fig. 9), but does not mimic climate projections, where the ensemble mean, and range are typically used. In the present setup, a future projection is treated as a deterministic prediction, rather than a probabilistic projection. Perhaps use of the climate 'pseudo-observed' run might be favoured over future predictions simply because there is less variability in the present climate? How sensitive are the results to taking the mean of all ensemble members minus the 'pseudo-reality' member (e.g. Fig. 3 in Räty et al. 2014)? This has the added benefit of involving much fewer permutations (and hence calculations).

*This is a good idea, which we have now implemented in our analysis suite. Results of this are included in the revised manuscript.*

4. The range of the projection matters. For example, Fig. 4 shows that there are future scenarios that exceed the present climate range. Hence, the worst-case 10-year precipitation event from the 'pseudo-obs' range would not include plausible future 10-year events. Therefore, more qualification is needed in the Abstract and Conclusions to guard against this possibility and the potentially misleading assertion that "the superior approach is to simply deduce future return levels from observations". Overall, the headline findings of the research could be presented in more nuanced ways, especially within the Abstract.

> *We are afraid that we do not understand the central statement of this point ("Hence, the worst-case …). Therefore, we are not able to comment on it.*

5. The Abstract and Introduction assert that "Severe precipitation events are usually projected using Regional Climate Model (RCM) scenario simulations." We gently remind the authors that statistical downscaling is also widely used for projecting severe precipitation events and suggest that more inclusive wording be used.

> *We agree that this suggestion is appropriate and have added a paragraph in the introduction (lines 68-74).*

---

## Author Response (AR2)

Dear editor,

We appreciate the very positive second report from referee #2 expressing content with our reactions to the issues in his first report.

Then the referee raises further issues. Our responses to these are in *italic* and section numbers and line numbers refer to the underlined marked-up version of the revised manuscript.

**Referee #1:**

I went through the authors response to my earlier comments and generally agree with the modifications they have made. The paper is more complete and better suited for publication.

My only other query is whether the authors noted any systematic changes in the relative errors for the two durations for the 12.5km runs compared to the 50km runs? This is an issue that comes up with RCM simulations often as the computational expense in the higher resolution runs is considerable. Any comments could be useful to others.

> *We analyse solely 12.5 km runs (EUR-11, line 203-204). Therefore, we are not able to give any qualified take on this query.*

Other than this, I noted down some specific issues in the order I read the paper. these are:

l65 - Authors should note the following two papers which are relevant here
Kim, Y., et al. (2020). "Impact of bias correction of regional climate model boundary conditions on the simulation of precipitation extremes." Climate Dynamics 55(11): 3507-3526.
Kim, S., et al. (2020). "Quantification of Uncertainty in Projections of Extreme Daily Precipitation." Earth and Space Science 7(8).
The first paper assesses the impact on extreme precipitation simulations once lateral and lower boundary biases are corrected. This is directly relevant to the present study. The second presents estimates of uncertainty in extreme precipitation simulations after bias correction for a range of models. I suggest the second one as it may give some guidance to authors to contrast their uncertainty estimates against those presented in the paper to gauge the extent of improvements made.

> *The first paper is a work where the forcing boundary conditions are bias-corrected. This is a different approach from ours.*

> *The second paper is it is about partition of variance between scenario, model and internal variability in GCMs. The paper by (Aalbers et al., 2018) is a more comprehensive study of the issue. We have added a sentence in the revised manuscript. (line 461)*

l72 and 73 - the referenced studies are not directly relevant to the present work because of the psuedoCC run. A bias corrected run would be a more appropriate comparison as noted in the above two papers.

*We are not able to identify the references of concern, since no references are given in these line numbers, neither in the raw nor in the annotated manuscript. Therefore, we are unable to react to this comment.*

l97 - I have long felt that in hydrological modelling settings, the issue of persistence is more important than simply correcting means.The Nesting Bias Correction papers and its variants in multivariate settings are my recommended alternatives where a water nalance simulation is needed. I urge the authors to exapnd their discussion about bias correction to mention this line of thought. The paper below given details of the software and how it should be used.
Mehrotra, R., et al. (2018). "A software toolkit for correcting systematic biases in climate model simulations." Environmental Modelling and Software 104: 130-152.

*We have added a short line and the reference (line 680)*

l205 - In my experience the fitting of the GPD creates instability in the shape and scale parameters, especially given the short record that are typically used. It would be good for the authors to discuss whether unstable parameter values resulted and whether the RE estimates are sensitive to such instability in a systematic manner.

*We are a bit in doubt about referee #2 really means l. 205. It is not specified whether it is referring to the marked-up version or not. In any case, extreme value analysis has not been mentioned at that point. We have added a sentence about uncertainties of parameters in EVA (lines 300-303)*

l328 - The 24h results show 3 cases where there is a decrease into the future. No such decrease is present in the hourly simulations. Is this a pattern that was common across the other regions the study focussed on? Would there be any reason for this? Perhaps linked to the instability in the GPD parameters?

*Line 328  is before any mentioning of results. Could the referee mean l. 428? A similar behaviour is seen in other regions. We have added a sentence about this (line 460-461)*

**Editor:**

In addition, a minor suggestion from my side - consider changing the label in the y-axis in Figure 4 to "Precipitation intensity [mm]"

*We think the term 'precipitation intensity' means a rate (mm /h). We will change the label to 'precipitation sum', which we use in this work (see line. Xx)*

**Other changes:**

Text have been adjusted for better readability.

Reference list format has been changed to conform to journal standard

[revised manuscript text omitted]